# CK2 and the Hallmarks of Cancer

**DOI:** 10.3390/biomedicines10081987

**Published:** 2022-08-16

**Authors:** May-Britt Firnau, Angela Brieger

**Affiliations:** Department of Internal Medicine I, Biomedical Research Laboratory, University Hospital Frankfurt, Theodor-Stern-Kai 7, 60590 Frankfurt, Germany

**Keywords:** Casein kinase 2, CK2, cancer, hallmarks of cancer

## Abstract

Cancer is a leading cause of death worldwide. Casein kinase 2 (CK2) is commonly dysregulated in cancer, impacting diverse molecular pathways. CK2 is a highly conserved serine/threonine kinase, constitutively active and ubiquitously expressed in eukaryotes. With over 500 known substrates and being estimated to be responsible for up to 10% of the human phosphoproteome, it is of significant importance. A broad spectrum of diverse types of cancer cells has been already shown to rely on disturbed CK2 levels for their survival. The hallmarks of cancer provide a rationale for understanding cancer’s common traits. They constitute the maintenance of proliferative signaling, evasion of growth suppressors, resisting cell death, enabling of replicative immortality, induction of angiogenesis, the activation of invasion and metastasis, as well as avoidance of immune destruction and dysregulation of cellular energetics. In this work, we have compiled evidence from the literature suggesting that CK2 modulates all hallmarks of cancer, thereby promoting oncogenesis and operating as a cancer driver by creating a cellular environment favorable to neoplasia.

## 1. Introduction

Casein kinase 2 (CK2) is a highly conserved serine/threonine kinase, constitutively active and ubiquitously expressed in eukaryotes [1]. CK2 can exist in a tetramer composed of two catalytic subunits, CK2α and/or CK2α′, associated with two regulatory subunits, CK2β. Importantly, each subunit is encoded by separate genes, *CSKN2A1* (CK2α), *CSKN2A2* (CK2α′), and *CSNK2B* (CK2β), and each can function independently of its association in the tetramer. Therefore, CK2α and CK2α′ maintain their kinase activity outside of their association with CK2β, which functions to modulate activity and confer substrate specificity [2]. CK2 phosphorylates serine/threonine residues within an acidic context (S/TXXD/E/pS/pT/pY) which can be found in a large variety of proteins in various subcellular compartments [3]. With over 500 known substrates and being estimated to be responsible for up to 10% of the human phosphoproteome [2,4], CK2 is of significant importance. Additionally, to having more than one-third of its substrates implicated in gene expression and protein synthesis [5], CK2 plays a critical role in development and differentiation [4], immunity [6,7], and DNA damage repair [8]. It also is essential for survival as CK2α and CK2β knockouts in mouse embryos are lethal [9,10]. A large number of CK2 substrates are involved in essential pathways of carcinogenesis [11]. Elevated levels of CK2 have been observed in malignant cells and are used to determine prognosis as well as survival in various forms of cancer e.g., breast, ovarian, prostate, lung, colon, and pancreatic cancer [12].

The hallmarks of cancer provide an organizational structure for the complexity and diversity of human tumor pathogenesis. They divide typical cancer traits into the maintenance of proliferative signaling, evasion of growth suppressors, resisting cell death, enabling of replicative immortality, induction of angiogenesis, the activation of invasion and metastasis as well as avoidance of immune destruction and dysregulation of cellular energetics [13]. It is important to acknowledge that CK2 phosphorylation sites have been shown to overlap with other posttranslational modifications (reviewed in [2]), thus participating in numerous regulatory events. Furthermore, it is known that CK2 activity is regulated in various biological processes and that CK2 interacts with different cellular proteins, leading to altered activity of CK2 as well as its partners (reviewed in [14]). However, this review will only focus on the distinctive roles of CK2 in cancer progression framed by the hallmarks of cancer.

## 2. Importance of CK2 in Different Hallmarks of Cancer

### 2.1. CK2 Is Involved in Selective Growth and Proliferative Advantage

A fundamental property of cancer cells is their ability to sustain chronic proliferation. With the deregulation of multiple cell regulatory systems, physiological cell growth and division cycle are altered thereby affecting tissue architecture and function. CK2 is involved at various points in this complex process.

#### 2.1.1. Modulation of Signaling Pathways

CK2 is known to have a role in healthy cell cycle control and progression [15]. One signaling pathway affected by CK2 and commonly dysregulated in cancer is the Wnt-signaling/β-catenin pathway (reviewed in [16]). The disturbance of this pathway has an impact on cellular proliferation in diverse human malignancies, including carcinogenesis of colorectal cancer (CRC), leukemia, melanoma, and breast cancer [16,17,18,19,20].

Wnt signaling consists of a group of signal transduction pathways, activated by the binding of a Wnt-protein ligand to a Frizzled family receptor, which passes the biological signal to the Disheveled (Dvl) protein inside the cell. So far, three different pathways have been characterized. The canonical pathway, implicated in the regulation of gene transcription, the noncanonical planar cell polarity pathway, involved in the regulation of the cytoskeleton as well as the noncanonical Wnt/calcium pathway which regulates calcium inside the cell [21]. The term “Wnt” is an amalgamation of wingless and Int1 [22].

CK2 affects the canonical Wnt-signaling/β-catenin pathway via phosphorylation at three points. By using wildtype (wt) mouse embryonal fibroblasts and human embryonic kidney cells (HEK293T) it has been shown that phosphorylation of Dvl in the basic region/PDZ region by CK2 led to its activation [23]. COS7, Wnt-1-C57MG cells, and mammary epithelial cells as well as Xenopus embryos were used to demonstrate that CK2 phosphorylates β-catenin in its armadillo repeat at Thr393, which protects β-catenin from proteasomal degradation and thereby promotes protein and co-transcriptional activity. This counteracts glycogen synthase kinase 3 β (GSK3β) phosphorylation of the N-terminus of β-catenin which promotes its degradation [24,25,26,27]. Furthermore, the phosphorylation of nuclear Lymphoid enhancer-binding factor 1 (LEF1) by CK2 significantly enhances its affinity for β-catenin and stimulates transactivation of the β- catenin/LEF1 complex [28]. Coupled with the fact that CK2 is overexpressed in many cancers, it is reasonable to assume that CK2 stimulates the transcription of cell proliferation promoting proteins such as Cyclin D1, Myc proto-oncogene protein (cMYC), monocarboxylate transporter 1 (MCT1), pyruvate dehydrogenase (acetyl-transferring) kinase (PDK) and fibronectin via the beforementioned pathways [29].

CK2 is further implicated in the modulation of the nuclear factor-kappa B (NF-κB) signaling pathway. The NF-κB signaling pathway regulates a variety of biological functions in the body, and its abnormal activation induces proliferation in different cancers [30,31,32,33,34,35]. It is tightly regulated, for example by the inhibition through classical IκB proteins (IκBα, IκBβ, and IκBε). After exposure to certain stimuli, the IκBα protein is degraded, which results in a decreased inhibitory effect on NF-κB. NF-κB in turn translocates from the cytoplasm to the nucleus, where it regulates the transcription of NF-κB target genes [36]. CK2 regulates all three IκB proteins [37]. It has been shown that phosphorylation by CK2 within the COOH-terminal proline (P), glutamic acid (E), serine (S), and threonine (T) (PEST) domain at the amino acid positions Ser283, Ser289, Thr291, and Ser293 of IκBα leads to its degradation [38]. This promotes the nuclear translocation of NF-kB and accelerates the expression of proliferative genes [39]. Moreover, CK2-mediated phosphorylation of IκBβ at Ser313 and Ser315 of the C-terminal PEST domain has been shown to be a prerequisite for NF-κB activation [40]. Some years ago, it has been discovered that inducible IKK (also named as IKK-i/IKKε) can lead to IκBα phosphorylation at Ser36 and NF-κB activation [41]. Although the significance of only one phosphorylation event at the NH2 terminus of IκBα is not yet entirely clear, it may predispose IκBα towards Ser32 phosphorylation and subsequent degradation [41]. Furthermore, CK2 has been implicated in the induction of IKK-i/IKKε as a signaling pathway in the aberrant activation of NF-κB in breast cancer [42]. The induction of IKK-i/IKKε has been recently shown in primary human breast cancers, rodent mammary tumors as well as cell lines. It could be demonstrated that the expression of IKK-i/IKKε correlated with the protein level of CK2α in mammary glands and breast tumors derived from mouse mammary tumor virus-CK2α transgenic mice [42]. Moreover, Ser529 of the NF-κB p65 subunit can be phosphorylated by CK2, with the effect of increasing NF-κB p65 transcriptional activity [43]. NF-κB signaling has been implicated in cellular senescence, a program of arrested proliferation and altered gene expression caused by different types of stress. Some cancer cells may adapt to high levels of oncogenic signaling by disabling their senescence- or apoptosis-inducing circuitry [13]. Using MCF-7 and HCT116 cells as well as Nematodes it has been established that CK2 acts as a key switch to regulate senescence-associated secretory phenotype (SASP) factors. The results suggest that nicotinamide adenine dinucleotide (NAD)-dependent protein deacetylase sirtuin-1 (SIRT1) connects CK2 down-regulation to SASP factors through NF-κB activation, which is mediated by both RelA/p65 deacetylation and activation of the protein kinase B (AKT)-IKK-IκB axis [44].

Extracellular signal-regulated kinases (ERKs) belong to the family of mitogen-activated protein kinases (MAPKs) and CK2 has been shown to contribute to their regulation. The function of ERKs and MAPKs is crucial for intracellular signal transduction networks which transmit signals from extracellular stimuli, such as growth factors, hormones, and neurotransmitters, thereby supporting cell proliferation [45,46]. CK2 has been identified as a Kinase Suppressor of Ras (KSR) KSR1 binding partner, which is required for KSR1 to facilitate ERK cascade signaling and contributes to the regulation of Raf kinase activity [47,48]. As demonstrated by Plotnikov et al., CK2 also directly controls the nuclear import of ERK by phosphorylating ERK at Ser244 and Ser246 [49,50]. While the phosphorylation of Ser246 has been shown to be sufficient for ERK nuclear translocation, the additional phosphorylation of Ser244 enhanced the kinetics of nuclear import [49,50]. Activated ERK stimulates transcriptional regulators ETS Transcription Factor ELK1 and c-Jun, leading to the expression of cell-cycle regulatory proteins such as D-type cyclins, thereby enabling the cell to progress through the G1 phase of the cell cycle [51]. In Serine/threonine-protein kinase B-raf mutant melanoma a scaffolding function of CK2 which promotes resistance to RAF- and MEK-targeted therapies has been demonstrated [52]. CK2 post-translationally regulates the ERK-specific phosphatase dual specificity phosphatase 6 (DUSP6), decreasing its abundance, and thereby maintaining the phosphorylation of ERK and promoting RAF-MEK kinase inhibitor resistance [52]. Surprisingly, this resistance did not rely on CK2 catalytic function but rather on the increase of KSR facilitation of ERK phosphorylation, elucidating a previously unknown mechanism of regulation [52].

CK2 has been identified as an interaction partner at different parts of the JAK-STAT signaling pathway. The JAK-STAT signaling pathway consists of Janus kinases (JAKs) that are signal transducers and activators of transcription proteins (STATs) and their respective receptors. Cytokines and growth factors bind to and activate the corresponding receptors, inducing a signaling cascade and initiating gene transcription [53,54,55]. The JAK-STAT signaling is involved in cell survival and cell proliferation and is often dysregulated in cancer [54,56]. CK2 is essential for the phosphorylation and activation of STAT3, thereby contributing to the survival of glioblastoma (GBM) cells [57]. In addition, CK2 is able to phosphorylate JAK2 on several tyrosine and serine residues, even though the biological function of those phosphorylation sites remains to be elucidated [58]. In line with this, Manni et al. demonstrated that the use of the CK2-inhibitor CX-4945 decreased the levels of phosphorylated STAT3 in multiple myeloma (MM) and mantel cell lymphoma [59]. This reduction of phosphorylation rendered MM cells irresponsive to Interleukin-6 (IL-6) elicitation. A suppression of *CCND1* and *IL-6*, STAT3 target genes, could be observed as well [59]. Inhibition of STAT1, STAT3, STAT5, and JAK2 phosphorylation and decreased expression of the suppressor of cytokine signaling 3 (SOCS3) was achieved by the use of small interfering RNAs or the CK2-inhibitor 4,5,6,7-tetrabromobenzotriazole (TBB) by Zheng et al. [58]. Overexpression of IL-6 has been reported in almost all types of tumors [60], where it can be responsible for the constitutive activation of the JAK/STAT signaling pathway [61]. Drygin et al. showed that CK2 is implicated in the regulation of IL-6 expression in a model of inflammatory breast cancer [62]. Using the CK2-inhibitor CX-4945 as well as siRNA against CK2 Drygin and coworkers detected an inhibited secretion of IL-6 in vitro as well as in vivo. This effect could be verified in a clinical trial with an inflammatory breast cancer patient [62].

Basically, two important phosphorylation sites of STAT3 are known–a tyrosine residue at amino acid position 705 (Tyr705) within the SH2 domain and a serine phosphorylation site at position 727 (Ser727) within the C-terminal transactivation domain [63]. However, CK2 seems to be only involved in the phosphorylation of position Tyr705. STAT3 has been found to be persistently activated in most human cancers mainly through its phosphorylation at Tyr705 [64]. In human glioma patients and in a rat orthotopic tumor model a negative correlation between CK2 and STAT3 phosphorylated at Ser727 has been demonstrated [64]. While overexpression of CK2 by transient transfection decreased STAT3 Ser727 phosphorylation, it was increased upon CK2 inhibition via Tetrabromocinnamic Acid (TBCA) and 2-(Dimethylamino)-4,5,6,7-tetrabromo-1H-benzimidazole (DMAT) [64]. Protein phosphatase 2A (PP2A), which is regulated by direct interaction with CK2α [65], acts as a negative regulator of STAT3 Ser727 phosphorylation. Cell lines stably overexpressing STAT3 Ser727A variant showed increased survival, proliferation, and invasion. In rat tumor models generated with the STAT3 S727A variant expressing tumor cell line were more aggressive, leading to the assumption that the CK2-PPA2 pathway regulates STAT3 Ser727 phosphorylation and herein tumorigenicity [64].

#### 2.1.2. Modulation of Transcription and Translation Factors

Another way to foster and maintain chronic proliferation is via the modulation of transcription and translation factors. One substrate of CK2 is Functioning forkhead box O3 (FOXO3a). FOXO3a binds to the promoter of apoptosis-inducing genes, such as Bcl-2-like protein 11 (Bim), Fas ligand (FasL), and tumor necrosis factor-related apoptosis-inducing ligand (TRAIL), and to the promoter of cell cycle inhibitors, such as Cyclin-dependent kinase inhibitor 1A (p21) and Cyclin-dependent kinase inhibitor 1B (p27) [66], thereby acting as a tumor suppressor. FOXO3a dysregulation, either by inactivation of the *FOXO3a* gene or cytoplasmic sequestration of the FOXO3a protein, is widely implicated in cellular proliferation in malignancy of breast, liver, colon, prostate, bladder, and nasopharyngeal cancers [67,68,69] (reviewed in [70]). CK2 is able to influence FOXO3a functioning via different pathways. CK2 promotes AKT signaling by phosphorylation of Phosphatase and TENsin homolog deleted on chromosome 10 (PTEN) on a cluster of serine and threonine residues (Ser380, Thr382, Ser385) in the C-terminal tail. This phosphorylation causes the stabilization of PTEN paralleled by a strong reduction of its phosphatase activity [71,72], and the phosphorylation of AKT on Ser129 by CK2 promotes AKT kinase activity [73]. This process leads to the nuclear export of FOXO3a during carcinogenesis [74]. In addition, a connection between CK2 and FOXO3a via Promyelocytic leukemia protein (PML) has been found in human prostate cancer [75]. Here, CK2 phosphorylates PML at Ser517 residue, promoting PML for proteasomal degradation, which fosters the continual aberrant activity of nuclear AKT. Active AKT phosphorylates FOXO3a expelling it out from the nucleus, promoting it for proteasomal degradation [75]. The proline-rich homeodomain protein/haematopoietically expressed homeobox protein (PRH/HHEX) is a transcription factor that controls cell proliferation, cell differentiation, and cell migration. The role of CK2-dependent phosphorylation of PRH in tumor development has been postulated [76,77,78]. CK2-dependent phosphorylation of PRH not only results in the inhibition of PRH DNA-binding activity, increased cleavage of PRH by the proteasome, and the dysregulation of PRH target genes [76,78] but also in increased cell proliferation and tumor cell migration and invasion as shown in prostate cancer cells [77].

The Apoptosis-Antagonizing Transcription Factor (AATF/Che-1) is an RNA polymerase II binding protein involved in several cellular processes, including apoptosis, response to stress, and proliferation, for example by sustaining global histone acetylation in MM cells [79]. CK2-mediated phosphorylation of Che-1 is required for its pro-proliferative activity. CK2 phosphorylates Che-1 at Ser316, Ser320 and Ser321, which enables Che-1/histone H3 binding [79].

CK2 has been implicated in eukaryotic translation. Eukaryotic translation initiation factor 5 (eIF5) is a 49 kDa protein and one of the most important proteins of translation initiation pathways. The overexpression of eIF5 has been associated with the induced translation of activating transcription factor 4 (ATF4) and possibly other genes with upstream open reading frames (uORFs) in their mRNA leaders through delayed re-initiation, thereby enhancing the survival of healthy and cancer cells under stress conditions [80]. Homma et al. was able to connect CK2 to eIF5 [81]. Using mass spectrometric analysis, Homma and coworkers presented that Ser389 and Ser390 of eIF5 are major sites of phosphorylation by CK2. eIF5 variants lacking CK2 sites had perturbed synchronous progression of cells through S to M phase, which resulted in a significant reduction in growth rate in COS-7 and HEK293 cells. This implicates CK2 in the regulation of cell cycle progression by associating with and phosphorylating a key molecule for translation initiation [81]. Since both CK2 and eIF5 are frequently overexpressed in different cancer types, their interaction might contribute to an accelerated cell cycle transition and thus to proliferation.

#### 2.1.3. Modulation of Regulatory Proteins

Epidermal growth factor receptor (EGFR) is a receptor tyrosine kinase commonly upregulated by various mechanisms in cancer, for example in pancreatic, breast, CRC, and non-small cell lung cancer (NSCLC) as well as GBM and head and neck cancer [82]. EGFR activates different pathways driving aberrant cell proliferation in cancer [82]. Constitutive activation of signaling pathways downstream of EGFR by CK2 contributes to the growth factor independence of cancers, which in turn drives cellular proliferation [56]. The aberrant activity of CK2 in the downstream signaling pathways often circumvents the effects of EGFR inhibitors, making dual inhibition of EGFR and CK2 signaling necessary. The success of simultaneous EGFR and CK2 inhibition has been shown in different studies. Bliesath et al. demonstrated that combined inhibition of EGFR with erlotinib and CK2 with CX-4945 augmented the attenuation of Phosphatidylinositol 3-kinase (PI3K)-AKT- mammalian target of rapamycin (mTOR) signaling, antiproliferative activity, and the killing of cancer cells in in vitro as well as in vivo models of NSCLC and squamous cell carcinoma [83]. Chou et al. detected that simultaneous blockade of interacting CK2 and EGFR pathways by tumor-targeting nanobioconjugates increased therapeutic efficacy against GBM multiforme with significant reductions in several signaling proteins important for tumor cell proliferation and invasion [84]. So et al. could show that autophagosome-mediated EGFR down-regulation induced by the CK2-inhibitor CX-4945 enhanced the efficacy of EGFR-inhibitors on EGFR-mutant lung cancer cells with resistance to the EGFR-inhibitors gefitinib and erlotinib and effectively inhibited cancer-cell proliferation [85]. Li et al. determined that the specific CK2-inhibitor Quinalizarin can reduce icotinib resistance in human lung adenocarcinoma cell lines and inhibit proliferation [86].

Progesterone is a 21-carbon steroid that exerts its primary physiological functions through binding to the progesterone receptors A and B (PR-A and PR-B), which initiate the transcription of targeted genes [87]. Especially PR-B has been implicated in the induction of proliferation in breast cancer [88]. An in vitro study demonstrated that CK2 phosphorylates human PR at Ser81, a residue located in the N-terminal region of PR unique to PR-B [89]. Using the estrogen-independent ER/PR positive T47Dco (T47D) variant cell line and HeLa-PR cells, Hagan et al. provided evidence that CK2-mediated phospho-Ser81 PR-B can drive the expression of genes contributing to breast cancer biology and a hyperproliferative state. This effect was most noticeable in the heightened expression of the anti-apoptosis protein Baculoviral IAP Repeat Containing 3 (BIRC3), of the 11-beta-hydroxysteroid dehydrogenase type 2 (HSD11β2), which inactivates the anti-proliferative effects of glucocorticoid receptor and of the Proheparin-binding EGF-like growth factor (HbEGF), a gene promoting mammary cell proliferation and breast cancer cell growth [90]. Since CK2 is frequently upregulated in breast cancer the phosphorylation of PR-B on Ser81 might additionally enhance tumor progression.

Cobb et al. found that CK2 phosphorylates Insulin-Like Growth Factor-Binding Protein-3 (IGFBP3) in prostate cancer at Ser167 and Ser175 [91]. While wt IGFBP-3- and IGFBP-3-S175A-induced apoptosis to a comparable extent, IGFBP-3-S167A was much more potent, showing that multisite phosphorylation can both negatively and positively impact the apoptotic potential of IGFBP3s [91]. IGFBP-3 is known to trigger cell proliferation and survival in breast cancer cell lines, since it is able to deliver IGFs to their receptors on the cell (IGF-dependent effects), even though IGF-independent effects on cell growth have been demonstrated as well [92]. However, it remains to be elucidated if the phosphorylation of IGFBP3 by CK2 plays a role regarding its influence on cell proliferation.

p21-activated kinase 1 (PAK1) has been implicated in various oncogenic pathways, inducing cytoskeletal remodeling, cell motility, promoting cell proliferation, regulating apoptosis, and accelerating mitotic abnormalities [93]. Shin et al. provided evidence that CK2 is critically involved in the activation of PAK1 [94]. CK2 phosphorylates PAK1 at Ser223 in vivo and in vitro, which is a key step for the activation and oncogenic conversion of PAK1. It has been demonstrated that overexpression of PAK1 is able to induce malignant transformation of prostate epithelial cells, while the growth of tumor cells could be abrogated by blockade of PAK1 Ser223 phosphorylation [94].

Another important substrate for CK2 is the heat shock protein 90 (HSP90) chaperone machinery, a key regulator of proteostasis under physiological and stress conditions in eukaryotic cells. HSP90′s main role is to assist other proteins in folding properly, stabilizing proteins against heat stress, and promoting protein degradation. Moreover, it regulates cell growth, proliferation in cancer cells, as well as stabilizes and/or activates networks of cancer facilitators [95,96,97]. CK2 drives proliferation via the phosphorylation of different signaling molecules involved with the HSP90 chaperone machinery.

HSP70-HSP90 Organizing Protein (HOP) is typically found in the cytosol, but it can shuttle between the nucleus and the cytoplasm due to the presence of a nuclear localization signal (NLS) at amino acids 222–239 (of mouse HOP). Specifically, phosphorylation of HOP by CK2 at amino acid position Ser189, contiguous to HOP NLS, regulates nuclear localization of HOP [98,99].

In addition to this, Muller et al. provided evidence that CK2 phosphorylated the C-termini of HSP90 and HSP70, in conjunction with Casein kinase 1 (CK1) and GSK3β [100]. The C-terminal phosphorylation of HSP90 and HSP70 prevented binding to the co-chaperone carboxy terminus of Hsc70 interacting protein (CHIP) and thus enhanced binding to HOP. Highly proliferative cells contained phosphorylated chaperones in complex with HOP and phospho-mimetic and non-phosphorylatable HSP variants showed that phosphorylation was directly associated with increased proliferation rate in various human cancer cell lines. Comparing primary tumor tissues with non-neoplastic tissue, predominantly C-terminally phosphorylated forms of HSP70 and HSP90 were measured, suggesting that tumors exhibit a dominant folding environment [100].

It is known that HOP promotes cell proliferation while GSK3β-mediated phosphorylation of lysine-specific demethylase 1 (LSD1), an epigenetic regulator, can contribute to the development of an aggressive cell phenotype [101]. Overexpression of LSD1 promoted tumor cell proliferation, migration, and invasion in NSCLC [102]. Working with different human cancer cell lines, Tsai et al. proposed that the HSP90–GSK3β complex translocates into the nucleus via the NLS of GSK3β while the nuclear import of HOP occurs after its phosphorylation by CK2. In the nucleus, HOP interacts with LSD1 which promotes its transfer to the HSP90–GSK3β complex, ultimately resulting in both appropriate LSD1 folding and subsequent phosphorylation [101]. Tsai and coworkers also demonstrated that HOP is required for GSK3β-mediated LSD1 phosphorylation, which promotes LSD1 stability and enhances cell proliferation [101]. Furthermore, it has been shown that CK2 phosphorylates the co-chaperone Cdc37 at Ser13 to enhance its interaction with HSP90, leading to increased activity of numerous kinases that are specifically chaperoned by Cdc37 and have pro-proliferative actions [103].

Bae et al. investigated the interplay of CK2 and the deletion in breast cancer 1 (DBC1) protein in gastric carcinomas [104]. The group reported that the inhibition of both CK2α and DBC1 decreased the proliferation and invasive activity of NCI-N87 as well as MKN-45 cancer cells. A mutation at the phosphorylation site Thr454 of DBC1 also downregulated the signals related to the epithelial-mesenchymal transition (EMT) [104]. DBC1 is a crucial endogenic inhibitor of SIRT1, which has been implicated in cancer progression and deacetylates p53, leading to a downregulation of its transcriptional activity and thereby to an enhancement of cell proliferation [105]. CK2-mediated phosphorylation increases the ability of SIRT1 to deacetylate p53 and protect cells from apoptosis after DNA damage [106]. Since CK2 is also known to inhibit p53 function in several ways [56], the exact role of CK2-mediated DBC1 phosphorylation in proliferation, invasiveness, and EMT remains to be further established.

Moreover, CK2 has been implicated in zinc signaling. Zinc is the second most abundant transition metal in the human body and dysfunctional zinc signaling is implicated in various disease processes including cancer [107,108,109]. Zinc transporter protein ZIP7 (ZIP7), a zinc influx transporter that is localized on the endoplasmic reticulum membrane [108], is post-translationally regulated by CK2-mediated phosphorylation at the amino acid residues Ser275 and Ser276 [110]. The phosphorylation of ZIP7 at these positions results in zinc release from intracellular stores, inhibiting protein tyrosine phosphatases (PTP), causing sustained activation of EGFR and Src [110] as well as ERK1/2 and AKT, which regulate signaling pathways leading to cancer cell proliferation and migration [111,112]. Like CK2, an aberrant expression of ZIP7 can be found in different cancers, for example in tamoxifen-resistant breast cancer, prostate, and CRC [110,113,114]. Zaman et al. demonstrated that knockdown of CK2α′ decreased the intracellular zinc level of breast cancer cells and in turn increased the cell viability while the opposite findings were obtained for prostate cancer cells. Knockdown of CK2β expression substantially increased the zinc level in breast cancer cell lines but decreased the zinc level in prostate cancer cells, implicating that different subunits of CK2 undertake different roles in the regulation of zinc homeostasis [115].

All CK2-dependent regulated proteins, which are described in this chapter and are involved in selective growth and proliferative advantage are summarized in Figure 1.

### 2.2. CK2 Facilitates Altered Stress Response Favoring Overall Survival

The apoptotic machinery consists of upstream regulators, receiving and processing extracellular death-inducing signals or detecting signals of intracellular origin, and downstream effector components. Attenuated apoptosis has been shown in tumors that manage to progress to high malignancy and are resistant to therapy [116]. The mechanisms of apoptosis involve signaling cascades, which can roughly be divided into the extrinsic or death receptor pathway and the intrinsic or mitochondrial pathway, even though molecules in one pathway can influence the other [117]. CK2 is involved in multifaceted ways in these apoptotic pathways.

#### 2.2.1. CK2 in the Extrinsic Apoptotic Pathway

##### Influence on Distinct Signal Transducers

CK2 can directly phosphorylate important signal transducers of the extrinsic apoptotic pathway and thus limit their functionality. The modulation of death receptor-mediated apoptosis by CK2 was shown in 2005 for the first time [118]. Wang et al. demonstrated in prostate cancer cells that overexpression of CK2 resulted in the suppression of apoptosis mediated by Tumor necrosis factor (TNF-α), TRAIL, and Fas-ligand (FasL) in cells responsive to these ligands, whereas downregulation of CK2 resulted in augmentation of apoptosis mediated by these ligands [118].

Although many types of cancers are sensitive to TRAIL-induced apoptosis, substantial numbers of cancer cells are resistant to TRAIL, especially highly malignant tumors such as pancreatic cancer, melanoma, and neuroblastoma [119]. Dolcet et al. provided evidence that TRAIL resistance in endometrial carcinoma cells is caused by elevated Cellular FADD-like IL-1β-converting enzyme (FLICE)-inhibitory protein (c-FLIP) levels [120] which are regulated by CK2 [121]. Llobet et al. demonstrated that an inhibition of CK2 by 5,6-dichlorobenzimidazole (DRB) correlated with the reduction of endogenous levels of c-FLIP, caused by simultaneous downregulation of c-FLIP expression and increased c-FLIP protein proteasomal degradation [121]. Forced expression of c-FLIP restored the resistance to TRAIL and Fas while knockdown of either Fas-associated protein with death domain (FADD) or caspase-8 abrogated CK2-inhibition triggered apoptosis. This implicates CK2 in the regulation of endometrial carcinoma cell sensitivity to TRAIL and Fas by regulating c-FLIP levels [121].

Vilmont et al. demonstrated an important role for CK2 in FADD nuclear localization [122]. They showed that CK2 phosphorylated FADD at Ser200 in acute monocytic leukemia cell line THP-1 and breast cancer cell line MCF10A. This phosphorylation was carried out by the CK2 holoenzyme in a CK2β-driven fashion. The model of Vilmont et al. proposed that CK1 association with and phosphorylation of FADD drives its nuclear localization followed by CK2 phosphorylation of FADD. This leads to sequestering of FADD in the nucleus and an inhibition of apoptosis by keeping FADD in the nucleus [122].

Furthermore, CK2 has been associated with the death-inducing signaling complex (DISC), a multi-protein complex formed by members of apoptosis-inducing cellular receptors, in rhabdomyosarcoma cells [123]. CK2-inhibition by DRB dramatically sensitized rhabdomyosarcoma cells to TRAIL-induced apoptosis while simultaneously inducing the rapid cleavage of caspase-8, -9, and -3. This inhibition also increased the level of recruitment of procaspase-8 to DISC and caspase-8-mediated cleavage of BH3-interacting domain death agonist (Bid). Bid in turn enhanced the release of proapoptotic factors such as cytochrome c and apoptosis-inducing factor (AIF) from the mitochondria with a subsequent degradation of X-linked inhibitor of apoptosis protein (XIAP). CK2 knockdown by shRNA in JR1 and Rh30 cells further substantiated the notion that CK2 regulates TRAIL signaling in rhabdomyosarcoma cells by modulating TRAIL-induced DISC formation and XIAP expression [123].

##### Protection from Caspase-Mediated Proteolysis

Another major mechanism underlying the CK2-mediated anti-apoptotic function is the protection of substrates from caspase-mediated proteolysis [124,125,126]. By using a peptide-based target screen Duncan et al. provided evidence that implicates CK2 in the global regulation of caspase signaling [126]. Kinase and in vitro caspase cleavage assays revealed that CK2 phosphorylates procaspase-3 at Thr174 and Ser176, which protects it from caspase-8- and caspase-9-mediated cleavage. They further showed that inhibition of CK2 by TBB enhanced the activation of caspase-3 during apoptosis as represented by increased abundance of cleaved poly-(ADP-ribose)-polymerase 1 (PARP1), a target of caspase-3. This led to the proposal of a model in which CK2 protects procaspase-3 from caspase-mediated cleavage, thereby inhibiting the activation of caspase-3 [126]. Turowec et al. further elaborated on the role of CK2 in caspase-3 signaling. They were able to show that CK2α′ phosphorylated caspase-3 in HeLa cells, thereby preventing its activation, which was negatively regulated by CK2β, suggesting that the asymmetric expression of CK2 subunits can differentially affect caspase activation and cancer cell survival [127].

Caspase-2 is the most evolutionarily conserved caspase and has been implicated in positive as well as negative regulatory functions in apoptosis [128,129,130]. Shin et al. detected a new role for CK2 in TRAIL-mediated apoptosis using the human esophageal cancer cell lines, TE2, HCE4, and HCE7, the human colon cancer cell lines, SW480 and HCT116 as well as the human malignant glioma cell line LN319 [131]. They were able to show that CK2 phosphorylates procaspase-2 at Ser157. CK2-inhibition by DRB led to the dephosphorylation, dimerization, and activation of procaspase-2. This enabled the procession of procaspase-8 to active caspase-8 thereby priming cancer cells for TRAIL-mediated apoptosis [131].

Of interest in the context of protection from caspase-mediated proteolysis is also the Hematopoietic lineage cell-specific protein 1 (HS1), a tyrosine multiphosphorylated protein, which is implicated in receptor-mediated apoptosis [132,133,134]. It has been shown in vivo in platelets and in vitro in Jurkat cells that CK2 phosphorylates HS1, which correlates with its implication in apoptosis, by conferring its resistance to caspase cleavage [132,133].

##### Influence on Inhibitors of Apoptosis and Growth Factors

Normally, apoptosis is tightly governed by inhibitors of apoptosis and growth factors supporting survival [135,136]. Apoptosis repressor with caspase recruitment domain (ARC) is a potent and multifunctional inhibitor of apoptosis [137]. CK2-mediated phosphorylation of ARC at Thr149 is a requirement for ARC to bind procaspase-8, associate with the outer mitochondrial membrane, and prevent apoptosis [138]. This phosphorylation has been shown to contribute to chemotherapy resistance by inhibiting doxorubicin (DOX) induced apoptosis [139]. In this study, Wang et al. detected that the CK2-inhibitors DRB and TBB were able to decrease the phosphorylation levels of ARC and sensitized human cervical cancer cells HeLa and human gastric cancer cell line SGC-7901 to apoptosis. In addition, synergistic effects of the combinatory treatment with DOX and CK2-inhibitors could be demonstrated in a tumor xenograft model [139].

Another substrate of CK2 and an important member of the inhibitor of apoptosis (IAP) family of proteins is Survivin. Its upregulation in a variety of human cancers is associated with high tumor grades, recurrence, and a poor clinical outcome [140,141]. Survivin inhibits TRAIL-induced apoptosis and the ratio between Survivin and TRAIL receptors is predictive of recurrent disease in neuroblastoma [142,143]. Using the colon cancer cells HT29, DLD-1, and SW480, the breast cancer cells ZR-75 and HEK293T cells Tapia et al. provided evidence that CK2 activity promotes survival by increasing Survivin expression via β-catenin-T cell factor (Tcf)/LEF-mediated transcription [144]. Fernández et al. demonstrated that even though CK2 inhibition reduced Survivin levels in MKN-45 as well as HEK293T cells and diminished β-catenin-Tcf/LEF-mediated transcription this effect could be rescued by Survivin overexpression. Since CK2 and Survivin are frequently overexpressed in cancer this mechanism could contribute to cell death resistance [145]. Barrett et al. demonstrated that CK2 phosphorylates Survivin specifically on Thr48 within its Baculovirus IAP Repeat domain [146]. Different in vitro analyses in HeLa and EM9 cells provided evidence that Thr48 is critical for Survivins ability to inhibit cell death. Mutation of Thr48 has been shown to alter the binding affinity of Survivin for borealin, a chromosomal passenger required for stability of the bipolar mitotic spindle [147]. Thus, CK2 is involved in the Survivin-dependent regulation of cell death, proliferation, and mitosis [146].

Furthermore, a connection of CK2 with the insulin-like growth factor (IGF)-binding protein 3 (IGFBP-3), which plays a key role in regulating the bioavailability and receptor interaction of the IGFs, has been found. IGFBP-3 is known to exert IGF-independent effects to inhibit cell proliferation and enhance apoptosis in many cell types [91]. Cobb et al. identified Ser167 to be phosphorylated by CK2 in the prostate cancer cell lines, LAPC4 and 22RV1, and the GBM cell lines, PC-3 cells, M059K and M059J. Using side-directed mutagenesis it could be demonstrated that IGFBP-3-S167A was much more potent to induce apoptosis due to its inability to undergo CK2 phosphorylation, which verifies CK2-mediated inhibition of apoptosis [91].

#### 2.2.2. CK2 in the Intrinsic Apoptotic Pathway

##### Influence on Tumor Suppressors and Distinct Signal Transducers

CK2 can also directly phosphorylate important signal transducers of the intrinsic apoptotic pathway, thus influencing their functionality and attenuating tumor suppressors. One very important tumor suppressor, PTEN, is mainly involved in the homeostatic maintenance of the PI3K/AKT cascade and its function is commonly lost in a large proportion of human cancers [148,149]. PTEN is able to induce apoptosis and cell cycle arrest through PI3K/AKT-dependent and -independent pathways [150]. CK2 and PTEN physically interact and CK2 mediates phosphorylation of PTEN in a cluster of residues, Ser370, Ser380, Thr382, Thr383, and Ser385, located in the C-terminal tail [151]. The phosphorylated tail binds to the phosphatase domain as a pseudosubstrate and to the C2 domain, preventing membrane binding of PTEN and leading to blockade of PTEN phosphatase activity [148,152,153,154]. Phosphorylation of the residues Ser370 and/or Ser385 inhibited the cleavage of PTEN by caspase-3 [152]. CK2 thus contributes to the dysfunction of PTEN in cancer and fosters inhibition of apoptosis.

Another substrate of CK2 is the tumor suppressor PML [155]. PML plays important roles in cell cycle regulation and survival; its inactivation or down-regulation is frequently found in cancer cells [156]. The PML protein is further implicated in apoptosis where it controls the functional cross-talk between ER and mitochondria at mitochondria-associated membranes [157]. Scaglioni et al. provided evidence that CK2 directly phosphorylates PML at Ser517, thereby promoting its ubiquitin-mediated degradation [158]. PML mutants resistant to CK2 phosphorylation displayed increased tumor-suppressive functions. Scaglioni and coworkers further uncovered an inverse correlation between CK2 kinase activity and PML protein levels in human lung cancer-derived cell lines and primary specimens [158].

p21, a cyclin-dependent kinase inhibitor, has been shown to interact with and bind to the CK2β subunit, inhibiting the activity of CK2 [159,160]. Moreover, CK2 has been demonstrated to phosphorylate p21 in vitro [161]. Localized in the cytoplasm, p21 binds to and inhibits the activity of proteins directly involved in the induction of apoptosis, including procaspase 3, caspase 8, and caspase 10,; it mediates the upregulation of genes encoding secreted factors with anti-apoptotic activities and suppresses the induction of pro-apoptotic genes by MYC and E2F1 [162,163]. Zhou et al. could show that AKT phosphorylates p21 at Thr145 resulting in its cytoplasmic localization [164]. This phosphorylation also enhances p21 protein stability [165]. Since it is known that CK2 phosphorylates and thereby aberrantly activates AKT [73] and AKT propels p21 antiapoptotic functions in the cytoplasm [163,164], this could be an important pathway to exert CK2′s antiapoptotic function. However, the ability of CK2 to directly phosphorylate p21 in vivo remains to be elucidated [161].

The Bcl-2 agonist of cell death (Bad) is a BH3-only member of the Bcl-2 family with important regulatory functions in apoptosis [166]. CK2 has been shown to phosphorylate Bad at Thr117 in cultured cortical neurons [167]. Bad has been closely linked to cancer and executes, depending on the phosphorylation state of three specific serine residues (Ser57, Ser99, and Ser118), pro- or anti-apoptotic functions [166]. The functional consequences of the CK2-mediated phosphorylation at Thr117 in cancer remain an interesting target for further research.

CK2 has also been implicated in the regulation of other Bcl-2 family members. Bid belongs to the Bcl-2 family of apoptotic regulators with a central role in integrating and converging signals at the mitochondria [168]. Stimulation of death receptors by their respective ligands leads to the activation of Bid and caspase-8 which cleaves Bid at Asp60 and Asp75 to generate tBid [169]. tBid relocates to mitochondria and promotes the release of apoptogenic factors [168]. Desagher et al. demonstrated that the cleavage of Bid by caspase-8 is regulated by CK1 and CK2 [170]. Using HeLa D98/AH2, MCF-Fas (stably Fas overexpressing MCF7S1 cells), Jurkat and PC12 cells Desagher and coworkers provided evidence that CK1 phosphorylates Ser61, Ser64, and Ser66, and that the presence of these phosphate groups C-terminal to Thr58 enables its phosphorylation by CK2. Once phosphorylated, Bid is insensitive to cleavage by caspase-8. Moreover, a variant of Bid that cannot be phosphorylated was found to be more toxic than wt Bid, thereby revealing a new mechanism for CK2-dependent inhibition of apoptosis [170]. Furthermore, Olsen et al. confirmed Thr58 as a phosphorylation site for CK2. They substantiated that Bid interacts with CK2α, and showed that the formation of tBid was reduced when Bid was phosphorylated by CK2 prior to caspase-8 cleavage [171]. Inhibition of CK2 by DRB or overexpression of CK2α demonstrated that the activity of CK2 uncoupled Bid cleavage from caspase-8 activation in individual HeLa cervical cancer cells, leading to the conclusion that CK2 provides transient tolerance to caspase-8 activities [169,172].

Additionally, CK2 has been shown to phosphorylate MYC-associated factor X (Max) in vivo at Ser11 thereby inhibiting the DNA-binding activity of Max homodimers but not of Myc/Max heterodimers [173,174]. Krippner-Heidenreich et al. demonstrated that CK2-mediated phosphorylation of Max at Ser11 inhibited the cleavage of Max by caspase-5 and thereby prevented apoptosis [175]. Max is the central component of the Myc/Max/Mad network of transcription factors that regulate growth, differentiation and apoptosis [173,175]. Max has been implicated as a tumor suppressor driver gene in MM [176] and was recently shown to function as a tumor suppressor and capable of rewiring the metabolism in small cell lung cancer (SCLC) [177].

The transcription factor Ikaros, the founding member of a zinc-finger protein family, has also been implicated in apoptosis [178]. Gurel et al. provided evidence that CK2 phosphorylates Ikaros in vitro at the positions Ser13, Thr23, Ser101, and Ser294 in VL3-3M2 cells and were able to confirm these phosphorylation sites in vivo [179]. Moreover, the group suggested after CK2-inhibition by DRB that these phosphorylations are dependent on the activity of CK2. They also observed that the phosphorylation of these sites is more sensitive to CK2-inhibition than other phosphoacceptor sites [179]. Popescu et al. further characterized the interplay between CK2 and Ikaros and demonstrated that phosphorylation of Ikaros by CK2 determines its ability to bind DNA, exerts cell cycle control, and its subcellular localization [180]. Furthermore, they reported that the dephosphorylation of CK2 phosphorylation sites by protein phosphatase 1 (PP1) stabilized the Ikaros protein thereby implicating CK2 in Ikaros destabilization [180].

Recently, it has been shown that Ikaros and CK2 regulate the expression of the mitochondrial transmembrane molecule B-cell lymphoma-extra large (Bcl-xl), encoded by the *BCL2L1* gene, which acts as an anti-apoptotic protein [181]. Hereby, CK2 modulates the chemosensitivity in high-risk B-cell acute lymphoblastic leukemia (B-ALL). Song et al. demonstrated that Ikaros regulated the expression of the *BCL2L1* gene by binding to the respective promoter, recruiting histone deacetylase 1 (HDAC1), and repressing *BCL2L1* expression via chromatin remodeling [181]. Phosphorylation of Ikaros by CK2 reduced Ikaros binding and recruitment of HDAC1 to the *BCL2L1* promoter, resulting in loss of repression of *BCL2L1* and increased expression of Bcl-xl with subsequent reduction of apoptosis. Inhibition of CK2 by CX-4945, in turn, increased binding of Ikaros to the *BCL2L1* promoter, additionally enhancing the sensitivity of B-ALL to DOX. Further experiments showed a beneficial synergistic effect of combinatorial treatment with CX-4945 and DOX in vitro and in preclinical models of high-risk B-ALL [181]. This implicates CK2-mediated phosphorylation in yet another pathway of apoptosis prevention.

CK2 has further been reported to phosphorylate Nucleophosmin/B23 (B23). B23 is a nucleolar phosphoprotein and its overexpression has been proposed as a marker in gastric, colon, ovarian, and prostate carcinomas [182]. B23 has been implicated in a number of pathways including apoptosis and genome stability and harbors protooncogenic as well as a tumor suppressor functions [183]. B23 is phosphorylated by CK2 at Ser125 in a highly acidic region during the interphase, leading to its nuclear localization [184]. Szebeni et al. suggested that CK2-mediated phosphorylation of B23 influences its molecular chaperone activity [185]. Wang et al. could show in prostate cancer cell lines ALVA-41 and PC-3 that CK2 and B23 colocalize in the nucleus after androgen administration [186]. Molecular and chemical downregulation of CK2 by siRNA and TBB resulted in the loss of nuclear-associated B23. Thus, Wang and coworkers suggested that the coordinate shuttling of B23 and CK2 in and out of the nucleus and their nuclear colocalization may represent an early event in their involvement in modulating responses to growth and apoptotic stimuli in the cell [186]. In this context, Perera et al. identified the impairment of CK2-mediated phosphorylation of B23 by CIGB-300 in vitro and in vivo [187]. They showed that ribosomal biogenesis, which was correlated with the rapid and massive onset of apoptosis in the SCLC line NCI-H82, was the first disturbed biological process [187]. Further investigation uncovered that the CK2-mediated phosphorylation of B23 was relevant in the modulation of a subset of genes involved in protein synthesis, mitochondrial ATP metabolism, and ribosomal biogenesis [188].

##### Counteracting p53-Apoptosis Inducing Functions

Another substrate of CK2 is the tumor suppressor Tumor protein p53 (p53), a central regulator in the apoptosis-inducing circuitry [189]. Over 50% of human cancers carry loss of function mutations in the *TP53* gene [190]. p53 limits cellular proliferation by inducing cell cycle arrest and apoptosis in response to cellular stresses such as DNA damage, hypoxia, and oncogene activation while transcriptionally regulating many apoptosis-related genes [191]. CK2 has been shown to directly phosphorylate p53 at Ser392 and CK2β interaction with p53 leads to reduced DNA binding and transactivation of p53 [192,193,194,195]. A common event during tumorigenesis is the mutation of the *TP53* gene leading to p53 variants which contribute to the development of tumors (reviewed in [196]). The conformation defective R175H p53 variant (p53H175) is one of the most frequent p53 variants, which exhibits dominant-negative activities over p53 and possesses gain of function properties [197]. Gillotin et al. investigated the ability of phosphorylation to modulate the functions and stability of mutant forms of p53 in H1299 and SaOS-2 cells [198]. They could validate that the mutation of CK2 phosphorylation site Ser392 specifically alters the stability of p53H175 and renders p53H175A392 more sensitive to E3 ubiquitin-protein ligase Mouse double minute 2 homolog (Mdm2)-mediated degradation than p53H175. This highlights the importance of this CK2 phosphorylation site for apoptosis as well as for the stability of p53 mutants [198].

The ubiquitination of p53 by E3 ubiquitin ligases targets p53 for proteasomal degradation, thereby contributing to the regulation of p53 homeostasis [199]. CK2 is able to influence the homeostasis of p53, albeit only indirectly. Ubiquitin carboxyl-terminal hydrolase 7 (USP7S) has been shown to be phosphorylated by CK2 at Ser18. This stabilizes USP7S and enhances the complex formation between USP7S and Mdm2. The formation of this complex in turn stabilizes Mdm2, which supports Mdm2-dependent p53 degradation [56,200]. This counteracts ATM-induced DNA-damage-elicited Mdm2 degradation and p53 stabilization [200].

Concerning CK2-dependent degradation of p53 the putative coiled-coil domain containing 106 protein (CCDC106) was identified as another p53-interacting partner by yeast two-hybrid screening [201]. CCDC106 has been demonstrated to enhance tumor growth and p53 degradation in a xenograft mouse model [202]. Its knockdown in cervical cancer HeLa and breast cancer MCF7 cells enhanced apoptosis by stabilizing p53 and suppressed cell viability, colony formation, migration, and invasion. CCDC106 overexpression exerted the opposite effects. CK2 has been shown to interact with CCDC106 via CK2β and phosphorylates CCDC106 at Ser130 and Ser147, which is a prerequisite for the interaction with p53 and nuclear localization. Amino acid substitution or CK2-inhibition by CX-4945 abrogated CCDC106-induced p53 degradation [202].

It has recently been demonstrated that head and neck squamous cell carcinoma with somatic mutations in *TP53* often retain and overexpress TAp73, a structural homologue of p53. TAp73 has the potential to replace the p53 function and represses the expression of key p53 target growth arrest and apoptotic genes [203,204]. Phosphorylation of TAp73 by CK2 inactivates this tumor suppressor and promotes the expression of cancer stem cell genes and phenotype [203,204].

Furthermore, CK2 has been linked to inositol hexakisphosphate kinase-2 (IP6K2). IP6K2, one of the major inositol pyrophosphate synthesizing enzymes, has been reported in physiologic mediation of cell death by binding to p53 [205]. This interaction decreases the expression of proarrest gene targets such as p21, thereby being required for p53-mediated apoptosis [206]. CK2 phosphorylates IP6K2 within its PEST sequence at Ser347 and Ser356, which enhances its ubiquitination and degradation, preventing the promotion of p53-mediated apoptosis [207]. Normally, IP6K2 generates IP7 which binds to CK2 leading to the stabilization of DNA-dependent protein kinase catalytic subunit (DNA-PKcs) and Ataxia Telangiectasia Mutated (ATM) Serine-protein kinase in a multistep process. DNA-PKc and ATM in turn phosphorylate and activate p53 at Ser15 [208]. CK2-mediated phosphorylation of IP6K2, therefore, interrupts the DNA-PK/ATM-p53 cell death pathway [207].

Moreover, deacetylase SIRT1 is a substrate of CK2. SIRT1 protects cells from stress-induced apoptosis by deacetylating a number of substrates including p53 [209,210,211]. CK2 has been reported to phosphorylate Ser659 and Ser661 of human SIRT1 in vitro and in vivo as well as Ser154, Ser649, Ser651, and Ser689 of mouse SIRT1 in vivo, after ionizing radiation (IR) [106,209]. Latter phosphorylation increased SIRT1s deacetylation rate and increased its substrate-binding affinity which led to the deacetylation of p53. This protected cells from apoptosis after IR-induced DNA damage [209]. The role of CK2 in SIRT1-phosphorylation was further elucidated by Jang et al. [212]. They showed that CK2-inhibition by DRB induced acetylation of p53 at Lys382 in HCT116 and HEK293 cells, which could be suppressed by SIRT1 activation. SIRT1 overexpression antagonized CK2 inhibition-mediated cellular senescence, a process thought to be an important tumor suppression process in vivo. Finally, maltose binding protein pull-down and yeast two-hybrid assays indicated that SIRT1 is bound to the regulatory CK2β subunit [212].

However, not only SIRT1 but also NAD-dependent protein deacetylase sirtuin-6 (SIRT6), which simultaneously plays a role as a tumor suppressor and oncogene, can be phosphorylated by CK2 [213,214,215]. Bae et al. characterized the relationship between CK2 and SIRT6 in breast carcinoma [216]. They showed that CK2 is able to phosphorylate SIRT6 at Ser338. Overexpression of SIRT6 increased proliferation while knockdown of SIRT6 and mutation at Ser338 decreased proliferation and invasiveness of MCF7 cells. This also downregulated the expression of matrix metallopeptidase 9, β-catenin, cyclin D1, and NF-κB. CK2-mediated phosphorylation of SIRT6 is therefore implicated in the progression of breast cancer [216]. Hussein et al. further elucidated the link between CK2 and SIRT6 and provided in vitro and in vivo evidence that the CK2-mediated phosphorylation of SIRT6 was associated with the induction of the DNA damage repair pathway proteins [217]. Hussein and coworkers suggested that this process potentiated the resistance to DOX thereby implicating the interplay of CK2 and SIRT6 in therapy and apoptosis resistance [217].

Beyond that, CK2 seems to participate in the regulation of Yin Yang 1 (YY1), a ubiquitously expressed multifunctional zinc finger transcription factor. YY1 is often overexpressed in cancers and is involved in the regulation of tumor cell growth, proliferation, migration, and metastasis [218,219]. It can directly bind to the promoter region of multiple long non-coding RNAs to regulate their expression, and has been shown to inhibit the activation of p53 in response to genotoxic stress [220], and to decrease p53 stability [218]. Riman et al. provided evidence that CK2α phosphorylates YY1 in vivo at Ser118 in its transactivation domain, proximal to a caspase 7 cleavage site thereby preventing YY1s cleavage during apoptosis [221]. Although the exact physiological consequences of this phosphorylation remain to be elucidated, the regulation of YY1 seems to be an additional way for CK2 to influence p53 and apoptosis.

Finally, the activity of OTU domain-containing ubiquitin aldehyde-binding protein 1 (OTUB1) has been suggested to be controlled by CK2. OTUB1 belongs to the ovarian tumor domain protease (OTU) subfamily of deubiquitinases and negatively regulates ubiquitination to control protein stability and activity [222]. OTUB1 is involved in a variety of cancer-related processes [222,223,224], including inhibition of apoptosis [225]. Herhaus et al. demonstrated that CK2 phosphorylates OTUB1 at Ser16, which led to the localization of OTUB1 in the nucleus, and was essential to repairing IR-induced DNA damage in osteosarcoma U2OS cells [226]. This suggests that CK2 may control OTUB1 activity by restricting its access to substrates in the cytosol or nucleus.

##### CK2 Modifies the Cellular Stress Response

The pervasive changes in cancer cells often lead to proteotoxic stress, which arises from alterations in various steps of protein synthesis and is attempted to be corrected by the cells’ control mechanisms [227]. CK2 has been implicated in the modification of cellular stress response, for example by its phosphorylation of Heat Shock Transcription Factor 1 (HSF1). HSF1 is activated in response to proteotoxic stress and enables cells to survive in suboptimal conditions. A high activity of HSF1 can be found in a wide range of tumors [228]. HSF1 is phosphorylated by CK2 at Thr142, which enhances its trimerization in the nucleus. The trimerization of HSF1 is necessary for its activator function and confers resistance to proteotoxic stress [229,230].

The unfolded protein response (UPR) is one of the pro-survival mechanisms triggered by the accumulation of unfolded or misfolded proteins in the endoplasmic reticulum (ER). It is used by cancer cells to bypass the apoptotic switch and exploited to promote proliferation and metastasis [231]. CK2 has been suggested to inhibit the branch of UPR that leads to apoptosis in MM [232]. Manni et al. provided evidence that the induction of ER stress by the cell permeable inhibitor thapsigargin increased CK2 activity in different MM cells. In contrast, CK2-inhibition by TBB promoted a reduction of the levels of ER stress sensors Serine/threonine-protein kinase/endoribonuclease IRE1 (IRE1) and Endoplasmic reticulum chaperone BiP (BIP/GRP78), increased phosphorylation of Eukaryotic translation initiation factor 2-alpha kinase 3 (PERK) and eukaryotic translation initiation factor 2 (eIF2α), and enhanced ER stress-induced apoptosis. A synergistic cytotoxic effect on MM cells in vitro and in vivo in mouse xenograft models was achieved by the combined inactivation of CK2 and HSP90 [232]. This can be explained by the fact that CK2, by phosphorylating Ser13 of Cdc37, tightens its association with chaperones, leading to MM plasma cell survival, proliferation, and enhanced stress-coping ability [233]. Furthermore, Muller et al. provided evidence that CK2 predominantly phosphorylates the C-termini of HSP70 and HSP90, which also supports that CK2 promotes a dominant folding environment in tumors [100].

Besides, Buontempo et al. linked CK2 with the UPR in T-cell acute lymphoblastic leukemia (T-ALL) [234]. Administration of CK2-inhibitor CX-4945 in vitro induced apoptosis in T-ALL cell lines as well as patient T lymphoblasts. It further downregulated PI3K/AKT/mTOR signaling in leukemic cells and led to a significant decrease of BIP/GRP78 with simultaneous upregulation of IRE1 and C/EBP-homologous protein (CHOP) [234]. Moreover, Hessenauer et al. could show that the inhibition of CK2 with TBB induced apoptosis via the ER stress response in prostate cancer cells, further supporting the significant influence CK2 exerts on the UPR to inhibit apoptosis [235]. In addition, ATF4, which is one of the main transcriptional effectors of UPR [236], is phosphorylated at Ser215 by CK2. Although a non-phosphorylatable variant of ATF4 was more stable than the ATF4 wt protein, the variant was less active at the promoters of the ATF4 targets *ATF3* and *CHOP*. This suggests an activating function of this modification [237]. CK2 also phosphorylates C/EBPδ and CHOP, two important players of UPR [238,239]. CK2-mediated phosphorylation of C/EBPδ at Ser57 increases its transcriptional activity [240], whereas the CK2-dependent phosphorylation of CHOP at Ser14–15 and Ser30–31 has an inhibitory effect [241]. By phosphorylating ATF4 and its binding partners, CK2 further seems to promote UPR. Recently Schmitt et al. demonstrated that CK2 phosphorylates cAMP-responsive element-binding protein 3 (CREB3)/Luman, a family member of ER resident transcription factors, at Ser46, thereby reducing the stability of CREB3 but not its transcriptional activity [242].

All CK2-dependent regulated proteins, which are described in this chapter and facilitate altered stress responses favoring overall survival are demonstrated in Figure 2.

### 2.3. CK2 Induces Angiogenesis and Vascularization

Under physiological conditions, angiogenesis is a highly regulated process and part of embryogenesis, wound healing as well as the menstrual cycle [243]. CK2 is involved in angiogenesis and neovascularization via various signaling pathways and in different cell types.

It has been shown that CK2 is able to regulate proteoglycan nerve/glial antigen (NG) 2 (NG2)-mediated angiogenic activity of human pericytes [244], a group of cells that have a leading function in supporting vascular sprouting and blood vessel formation [245]. Pharmacological inhibition of CK2 by CX-4945 suppressed pericyte proliferation, migration, spheroid sprouting and the stabilization of endothelial tubes. In vivo, implanted Matrigel plugs containing CX-4945-treated pericytes exhibited a lower microvessel density when compared to controls [244].

Concerning neovascularization, two CK2 inhibitors, emodin, and TBB, have been identified to stabilize retinal endothelial cell tubes on the basement membrane matrix, inhibit growth-factor-stimulated endothelial cell migration, proliferation, and secondary sprouting on this matrix, and significantly decrease oxygen-induced retinal neovascularization in mice [246]. In addition, it has been shown that bone marrow-derived human stem cells injected into the vitreous of neonatal mice could incorporate into the retinal neovasculature. This process was significantly reduced by CK2 inhibitors emodin, DRB, 4,5,6,7-tetrabromobenzotriazole (TBBt) and 4,5,6,7-tetrabromobenzimidazole (TBBz) [247]. Based on this, it has been suggested that CK2 may interfere with human stem cell recruitment during angiogenesis.

Moreover, the inhibition of CK2 in primary human retinal pigment epithelial (RPE) cells resulted in a significant inhibition of the upregulation of the vascular endothelial growth factor (VEGF) by oxidized phospholipids [248], which implicates that CK2 seems to act via VEGF in the process of angiogenesis.

Recently, it has been detected in endometric lesions and myeloid leukemia that phosphorylation of PRH by CK2 inhibits its DNA-binding activity. PRH represses the transcription of multiple genes encoding components of the VEGF-signaling pathway and thereby influencing angiogenesis [78,249] which is abrogated by CK2 phosphorylation. Phosphorylation of PRH by CK2 also decreased the nuclear association of PRH and induced its cleavage by the proteasome in K-562 cells [78].

Another mechanism by which CK2 influences VEGF is a positive feedback loop connecting Survivin expression in murine as well as human tumor cells to PI3K/AKT enhanced Tcf/LEF-dependent transcription followed by secretion of VEGF and angiogenesis [144,145]. CK2 kinase activity promotes survival by increasing Survivin expression. Even though CK2 inhibition reduced Survivin levels in gastric adenocarcinoma cells as well as in HEK293 cells and diminished β-catenin-Tcf/LEF-mediated transcription this effect could be rescued by the overexpression of Survivin alone. Since CK2 and Survivin are frequently overexpressed in cancer this mechanism might contribute to increased angiogenesis [145].

VEGF is a key regulator of vascular permeability (VP) which allows the free, bidirectional passage of small molecules and gases as well as plasma proteins [250]. Typically, VEGF mediates VP via activating downstream signaling factors such as Src kinase [251] which has been detected to be over-expressed and highly activated in a wide variety of human cancers [252]. Phosphorylation of the protein tyrosine phosphatase DEP-1 on Tyr1311/Tyr1320 mediates the activation of Src, and promotes Src-dependent angiogenic responses including endothelial cell permeability. DEP-1 Thr1318 is part of a CK2 consensus phosphorylation site and has been identified as a CK2 substrate. Modulation of CK2 expression or activity in endothelial cells regulated Thr1318 phosphorylation, and correlated with the status of Tyr1320 phosphorylation, Src activation, and cell permeability [253]. CK2-dependent phosphorylation of DEP-1 on Thr1318 is therefore assumed to be part of a regulatory mechanism that channels the activity of DEP-1 towards Src, allowing its optimal activation and the promotion of endothelial cell permeability [254].

Angiogenesis can also be regulated via the PML tumor suppressor-AKT/mTOR pathway [255]. PML can recruit PP2A which dephosphorylates and thereby inactivates AKT [253]. PML physically interacts with mTOR and negatively regulates its association with its activator Rheb by favoring mTOR nuclear accumulation [256]. CK2 regulates PML protein levels by phosphorylation at residue Ser517, leading to its ubiquitin-mediated degradation [255,257,258]. Through negatively regulating the AKT-mTOR pathway PML inhibits angiogenesis. PML deficiency leads to increased neoangiogenesis and elevated expression of pro-angiogenic factors such as Hypoxia-inducible factor 1 *alpha* (HIF-1α) and VEGF in human and mouse tumors [255,256].

Finally, CK2 is an important regulator of HIF-1α [259], a transcription factor that regulates the expression of secreted factors that mediate the angiogenic phenotype in most cancers [260]. The HIF-1α-dependent activation of VEGF transcription in hypoxic cells [261] and its upregulation of VEGF is required to promote the angiogenic phenotype for example in uveal melanoma [262]. The regulation of HIF-1α by CK2 happens partially via phosphorylation of the von Hippel–Lindau protein (VHL) which is the substrate recognition component of an E3 ubiquitin ligase and functions as a master regulator of HIF activity by targeting the hydroxylated HIF-1α subunit for ubiquitylation and rapid proteasomal degradation under normoxic conditions [263]. CK2 can downregulate VHL expression at the transcriptional level by phosphorylating HDAC1 and HDAC2 [264] as well as post-translationally by destabilizing phosphorylation in the NH2-terminal acidic domain of VHL [265]. The CK2-induced destabilization of VHL also results in p53 inactivation, leading to the abolition of HIF-1α transcription inhibition [265,266]. Another mechanism that induces HIF-1α accumulation is the phosphorylation of Cdc37 by CK2. This allows HSP90/Cdc37 dimer formation and the subsequent interaction of this complex with HIF-1α, which is essential for its cellular stability [267,268].

All CK2-dependent regulated proteins, which are described in this chapter and are associated with angiogenesis and vascularization are shown in Figure 3.

### 2.4. CK2 Promotes Invasion and Metastasis

Invasion and metastasis are a multi-step process where each step is potentially rate limiting. Consecutive biological changes define each of these steps, starting with local invasion, intravasation into nearby blood and lymphatic vessels, with subsequent transport of cancer cells into more distant tissues, adherence, extravasation into organ parenchyma, formation of micrometastases and finally growth of these micrometastases into macroscopic tumors [269]. Dysregulated CK2 levels occur in a variety of cancers and have been associated with a more aggressive spreading-prone cancer cell behavior [270,271]. CK2 is able to alter signaling mechanisms that drive tumor invasion and metastasis [259,266,272,273].

#### 2.4.1. Cell Adhesion

A common alteration of carcinoma cells is the loss of E-cadherin, a key cell-to-cell adhesion molecule that forms adherens junctions with adjacent epithelial cells, helps to assemble epithelial cell sheets and maintains the quiescence of cells within these sheets [274]. N-cadherin, which is normally expressed in migrating neurons and mesenchymal cells during organogenesis, is upregulated in many invasive carcinoma cells [275]. Zou et al. demonstrated that CK2α is overexpressed in CRC and modulates cell proliferation and invasion via regulating EMT-related genes. Knockdown of CK2α suppressed cell motility and invasion and increased the expression of E-cadherin [276]. Ko et al. provided evidence in the esophageal carcinoma cell lines, HCE4 and TE2, that high CK2 activity could switch cadherin expression from type E to N, which correlated with increased invasiveness [277]. Myeloid zinc finger 1 (MZF1) is an N-cadherin transcription factor. CK2 phosphorylates MZF1 at Ser27 in human esophageal cancer cell lines and HEK293 cells, thereby stabilizing MZF1 and inducing N-cadherin transcription [278].

Microtubule-associated protein RP/EB family member 2 (MAPRE2) is a member of the microtubule-binding EB1 protein family, which interacts with adenomatous polyposis coli protein (APC), a key regulatory molecule in the Wnt signaling pathway. MAPRE2 is highly expressed in pancreatic cancer cells and is associated with increased perineural invasiveness, poor outcome, and prognosis [279]. Stenner et al. demonstrated in HEK293 cells that CK2 interacts with and phosphorylates MAPRE2 at Ser236. Stable expression of MAPRE2 led to a significant cleavage and downregulation of N-cadherin and impaired adhesion [280]. Abiatari et al. provided evidence that pancreatic cancer cells expressing a phospho-mimicking MAPRE2-ASP236 variant show a marked decrease of adhesion to endothelial cells under shear stress [279]. The downregulation of endogenous MAPRE2 under shear stress has been speculated to improve cellular adhesion [279]. Thus, the phosphorylation of MAPRE2 by CK2 seems to trigger cancer cell adhesion.

Another molecular mechanism linking CK2 with cellular adhesion is the phosphorylation of Vitronectin (Vn), an adhesive glycoprotein found in the extracellular matrix (ECM) of various cells. Vn is recognized by a family of receptors known as integrins, and promotes cell attachment, spreading, and migration [281]. Vn is phosphorylated by CK2 at Thr50 and Thr57, which significantly enhanced the adhesion and spreading of bovine aorta endothelial cells [281]. Seger et al. could show that this effect is mediated via the α_v_β_3_, not via the α_v_β_5_ integrin in HeLa cells and NSCLC cell line H1299 [282]. The extent of AKT activation coincided with the enhanced adhesion and AKT activation, as well as elevation of adhesion, were PI3K-dependent in these cells [282]. This further supports the significant influence of CK2 on cellular adhesion.

The disruption of tight junctions occurs during the detachment of the tumor cell from the primary tumor, the intravasation through the endothelium, and the extravasation by the circulating tumor cell [283]. Occludin is a necessary integral protein for tight junction structure and function which is implicated in bone metastasis, and lung and breast cancer [284,285,286]. The C-terminal cytoplasmic tail of human occludin regulates the assembly/disassembly and the barrier properties of tight junctions and the inhibition of CK2-mediated phosphorylation of occludin at position Ser408 (by TBB, DMAT, and EMD) has been shown to elevate transepithelial resistance by reducing paracellular cation flux and reverse IL-13–induced, claudin-2–dependent barrier loss [287,288].

#### 2.4.2. Disturbance of mRNA Translation

One process commonly disturbed and therefore contributing to cancer cell spread and invasion is the deregulation of messenger RNA (mRNA) translation [289]. eIFs assist to stabilize the formation of the functional ribosome around the start codon and provide regulatory mechanisms in the translation initiation of mRNA. Dysregulated mRNA translation and disturbed expression of eIFs foster oncogenic progression [80]. CK2 has been shown to be involved in the modification of eIFs. One very important eIF is eIF4E, which is elevated and directly related to disease progression in multiple human cancers. Its overexpression or hyperactivation is known to drive cellular transformation and malignant progression in experimental models [290]. Ye et al. demonstrated that both extracellular-signal regulated kinases (ERK) and AKT signaling are required to activate eIF4E-initiated cap-dependent translation via the regulation of translational repressor 4E-binding protein 1 (4E-BP1), which maintains CRC transformation, motility, and metastasis [291].

The eIF4E protein forms, together with the ATP-dependent helicase eIF4A and the scaffolding protein eIF4G, the eIF4F complex. Gandin et al. demonstrated the importance of CK2 for the regulation of this eIF4F complex assembly [292]. Hereby, CK2 stimulated the phosphorylation of eIF2β and increased eIF4F complex assembly via the mTOR complex 1 (mTORC1)/4E-BP pathway which promoted cell proliferation [292]. Furthermore, the eIF4F complex assembly is driven by CK2-mediated phosphorylation of PTEN and AKT [71,72,73]. Although all capped mRNAs require eIF4F for translation, mRNAs encoding for proteins involved in tumor invasion and metastasis (Matrix metalloproteinases 9 (MMP-9), heparanase) are exceptionally dependent on elevated eIF4F activity for translation [290]. CK2 thereby is able to influence dysregulated mRNA translation directly or indirectly e.g., via activation of ERK [49,293] and AKT [73].

Serine/threonine-protein kinase RIO1 (RIO1) is involved in the final steps of cytoplasmic maturation of the 40S ribosomal subunit. A connection between RIO1 and CK2 has been firstly described in yeast where the phosphorylation of CK2-modified serines was essential for the complete activity of RIO1 and the lack of RIO1 phosphorylation has been shown to be disadvantageous for cell proliferation [294,295]. Moreover, RIO1 has been demonstrated to be overexpressed in different subtypes of human lung and breast cancer. Overexpression of RIO1 activated NF-κB signaling and promoted cell cycle progression, lung colonization in vivo, and knockdown of RIO1 in colon, breast, and lung cancer cell lines strongly impaired proliferation and invasiveness in conventional and 3D culture systems [296]. RIO1 is methylated at Lys411 by Histone-lysine N-methyltransferase SETD7 methyltransferase which enables F-box protein 6 (FBXO6) to interact with RIO1 and induce its ubiquitination. RIO1 methylation reduced the tumor growth and metastasis in a mice model whereas CK2 phosphorylation of RIO1 at Thr410 antagonized Lys411 methylation, stabilized RIO1 and impeded the recruitment of FBXO6 to RIO1 thereby driving colorectal and gastric cancer development [297].

#### 2.4.3. Disruption of Receptors and Signaling Pathways

The disruption of receptors and their related signaling pathways can promote invasion and metastasis which has been shown for example in prostate cancer (PCa). PCa is the most commonly diagnosed cancer among men in western countries. Androgen receptor (AR) signaling plays key roles in the development of PCa [298]. CK2 and AR are closely linked to the pathogenesis of PCa. Trembley et al. showed that the inhibition of CK2 led to a downregulation of the AR-dependent transcription and a significant decrease of the AR protein in vitro as well as in vivo [299]. In contrast, elevated CK2 levels could be associated with increased levels of AR and NF-κB p65 in non-transformed prostate cells as well as androgen-responsive and castration-resistant malignant prostate cells. Also, AR and CK2α RNA expression were strongly correlated [299]. The AR promotes prostate cancer metastasis via the induction of EMT with Snail activation and upregulation of eIF5A2. Only a low content of AR seems to be required to induce EMT phenotype as an inverse correlation between expression levels of AR and androgen-mediated EMT in prostate tumor epithelial cells has been demonstrated [300,301]. CK2′s role in the induction of EMT thereby seems to be mediated by the maintenance and promotion of AR.

Furthermore, the nuclear receptor corepressor (NCoR), a critical transcriptional repressor of nuclear receptors, significantly influences the process of cancer cell invasion. CK2α phosphorylates the C-terminal domain of NCoR at Ser2436, which stabilizes NCoR and avoids its ubiquitin-dependent proteasomal degradation [302]. Increased phosphorylation is inversely correlated with the mRNA level of interferon-γ-inducible protein 10 (IP-10) which is regulated by NCoR. CK2 inhibition abrogated NCoR phosphorylation, IP-10 transcriptional repression, and the invasion activity of prostate cancer cells PC-3 [302]. Similar effects were demonstrated in human esophageal cancer cells further linking CK2 with the promotion of invasion [303].

Of importance concerning the inhibition of cancer metastasis is the Breast cancer metastasis suppressor 1 (BRMS1). BRMS1 is decreased in NSCLC and its loss correlates with increased metastases [304]. Liu et al. provided evidence that CK2α’ drives lung cancer metastasis by targeting BRMS1 nuclear export and degradation via the phosphorylation of Ser30 [304]. Mutation of Ser30 in BRMS1 or CK2 inhibition with CX-4945 abrogated cell migration, invasion, and decreased NSCLC metastasis by 60-fold. The analysis of human NSCLC specimens confirmed that CK2α’ and cytoplasmic BRMS1 expression levels in cancer tissues are associated with increased tumor recurrence, metastatic foci, and reduced disease-free survival [304].

In addition, CK2 participates in the regulation of the Notch signal transduction pathway, which mediates important cellular functions through direct cell-to-cell contact and consists of four family members (Notch1-4). Deregulated expression of Notch receptors and ligands is observed in a variety of solid tumors, including cervical, skin, pancreatic, ovarian, lung, prostate, and breast carcinomas. Notch signaling is implicated in all hallmarks of cancer, either in an oncogenic or tumor suppressive role [305,306]. Ranganathan et al. identified the Notch intracellular domain as a novel target of phosphorylation by CK2 [307]. Phosphorylation of Ser1901 by CK2 appeared to generate a second phosphorylation site at Thr1898, combined phosphorylation resulted in decreased DNA binding and lower transcriptional activity. However, it is unclear if the detected phosphorylation is specific for one of the four Notch receptors or ubiquitous [307]. In contrast, other groups were able to link CK2 and Notch1 signaling with increased invasion and metastasis in breast, gastric and hepatocellular cancer [308,309,310]. In lung cancer cell lines A549 and H1299 CK2α has been detected as a positive regulator of Notch1 signaling and inhibition of CK2α down-regulated Notch1 signaling, subsequently reducing a cancer stem-like cell population [311]. Moreover, Lian et al. postulated that combined inhibition of CK2 by CX-4945 and a gamma secretase inhibitor can destroy the stability of Notch1 and reduce the growth and survival of human T-acute lymphoblastic leukemia cells [312]. However, it remains unclear whether the cytotoxic effect of CX-4945 on T-ALL cells has been directly associated with the repression of Notch1 signaling or not [312]. Since both CK2 and Notch play a role in a variety of signaling pathways, the exact influence of CK2 on Notch1 needs to be further investigated.

Besides, transforming growth factor β (TGFβ) is overexpressed in advanced cancers and promotes tumorigenesis by inducing EMT, which enhances invasiveness and metastasis [313]. TGFβ signaling is very complex and contains a large number of proteins, including CK2. In A549 and HEK293 cells, TGFβ signaling has been shown to decrease CK2β but not CK2α protein levels, resulting in a quantitative imbalance between the catalytic α and regulatory β subunits and leading to an increase in CK2 activity [314]. The decrease in CK2β expression has been shown to be dependent on TGFβ receptor (TGFBR) I kinase activity and the ubiquitin–proteasome pathway. Taken together this provides evidence that CK2 activation is required for TGFβ-induced EMT [314].

PRH/HHEX is a transcription factor that controls cell proliferation, cell differentiation, and cell migration. PRH downregulates cell migration, but is also part of a feedback loop involved in activating transcription of Endoglin, TGFβ1 co-receptor. TGFβ1 in turn down-regulates PRH expression in prostate cells and thereby up-regulates cell migration, enabling a more precise control of cell behavior [315]. CK2-dependent phosphorylation of PRH not only results in the inhibition of PRH DNA-binding activity, increased cleavage of PRH by the proteasome, and the dysregulation of PRH target genes [76,78] but also in an increase in cell proliferation and tumor cell migration and invasion as shown in prostate cancer cells [77,316].

As already mentioned in “CK2 is involved in selective growth and proliferative advantage”, CK2 is closely linked to Wnt/β-catenin signaling, which is frequently disturbed in cancer [16]. Ji et al. demonstrated that EGFR activation resulted in disruption of the complex of α-catenin and β-catenin, thereby abrogating the inhibitory effect of α-catenin on β-catenin transactivation via CK2α-dependent phosphorylation of α-catenin at Ser641 [317]. This activation cascade has been shown to mediate tumor cell invasion. The levels of α-catenin Ser641 phosphorylation in human GBM specimens correlated with levels of ERK1/2 activity, which has been linked to glioma malignancy [317]. CK2 also phosphorylates β-catenin in its armadillo repeat at position Thr393, which protects β-catenin from proteasomal degradation and promotes β-catenin’s protein and co-transcriptional activity as shown in COS7, Wnt-1-C57MG cells and mammary epithelial cells as well as Xenopus embryos [24,25,26,27].

Moreover, β-catenin activation has been shown to downregulate cell-cell junction-related genes and induce EMT in CRCs, contributing to CRC aggressiveness [318]. This effect is in part due to the binding of β-catenin to T-cell and lymphoid enhancer 4 (TCF4), initiating the subsequent binding to the Zinc finger E-box-binding homeobox 1 (Zeb1) whose expression promotes tumorigenesis and metastasis in carcinomas [319]. Additionally, β-catenin induced transcriptional expression of Programmed cell death ligand 1 (PD-L1) which promoted GBM immune evasion [320]. Since PD-L1 is a promising target for a new oncological therapy, it is exceedingly interesting that CK2 is also involved in the regulation of PD-L1. PD-L1 has been implicated in the influence on cell spreading, migration, and invasion for example in head and neck cancer cells [321]. Recently it has been demonstrated that the CK2-dependent phosphorylation of PD-L1 at positions Thr285 and Thr290 disrupted PD-L1 binding with speckle-type POZ protein and protected PD-L1 from cullin 3 ubiquitin E3 ligase complex-mediated proteasomal degradation in cancer and dendritic cells (DC). Inhibition of CK2 decreased PD-L1 protein levels by promoting its degradation [322]. The relevance of this will be further elucidated in “CK2 in genome instability and mutation”.

In addition, the endothelin-1 (ET-1) axis contributes to the pathophysiology of several cancers by promoting tumor development and progression. Zinc metalloprotease endothelin-converting enzyme-1 (ECE-1), which has four isoforms, ECE-1a-d, has been found to be aberrantly expressed in several tumors and cancer cell lines. Especially ECE-1c contributes to cancer aggressiveness and progression [323]. Different groups provided evidence that CK2 phosphorylates the N-terminus of ECE-1c at positions Ser18 and Ser20, which enhances its stability, and promotes invasiveness and aggressiveness of CRC cells [324,325,326].

Zinc is the second most abundant transition metal in the human body [107]. Recently it has been shown that zinc induced EMT in human lung cancer H460 cells via a superoxide anion-dependent mechanism. The treatment of cells with zinc significantly increased EMT markers N-cadherin, vimentin, Snail, and Slug and decreased E-cadherin, simultaneously showing increased cancer cell motility [327]. CK2 could further contribute to this by its phosphorylation of zinc transporter ZIP7 on residues Ser275 and Ser276 [110], which results in zinc release from intracellular stores, and activates several pathways driving cancer cell proliferation and migration [111,112]. Taken together, this shows the significant and complex role of CK2 in driving EMT, invasion, and metastasis.

#### 2.4.4. Dysregulation of Proteins Normally Relevant in Embryogenesis

Similarities between embryogenesis and cancer progression have been suggested some years ago [328]. The dysregulation of proteins normally involved in embryogenesis could therefore foster tumorigenesis. Hedgehog (Hh) signaling is required for cell differentiation and organ formation during embryogenesis [329]. Hh dysregulation has been associated with basal-cell carcinoma, medulloblastoma, SCLC, NSCLC, pancreatic adenocarcinoma as well as CRC [330,331,332]. CK2 has been reported to positively regulate Hh signaling in A549 and H1299 lung cancer cell lines. Zhang et al. provided evidence that CK2α inhibition decreased downstream effector in Hh signaling Zinc finger protein Glioma-Associated Oncogene 1 (GLI1) expression and transcriptional activity and enhanced its degradation [333]. In hepatocellular carcinoma Hep G2 cells an inhibition of migration and invasion as well as a reduction in the expression of GLI1 and PTCH1 after CK2α inhibition could be demonstrated [334]. Human mesothelioma samples showed a positive correlation between GLI1 and CK2α expression. CK2α genetic silencing or pharmacological inhibition with CX-4945 reduced the expression and transcriptional activity of GLI1 [335]. Phosphoproteomics identified CK2 as a driver of Hh signaling and a therapeutic target in medulloblastoma [336]. Here, CK2 has been detected to be critical for the stabilization and activity of the transcription factor GLI2, a downstream effector in Hh signaling.

Moreover, the transcription factor Twist is an important regulator of the cranial suture during embryogenesis, inducing EMT with a distinct increase in cell motility. Aberrant EMT triggered by Twist in human mammary tumor cells has been first reported to drive metastasis to the lung in a metastatic breast cancer model [337]. CK2 has been shown to enhance IL-6 expression in inflammatory breast cancer [62] and has been previously implicated to be involved in Twist mediated EMT [338]. Twist has been determined to be stabilized by IL-6 and IL-6 enhanced tumor cell motility could be demonstrated in head and neck cancer cells through activation of CK2 [338].

#### 2.4.5. Extracellular Matrix

The destruction of ECM is associated with cancer cell invasion [339]. Furthermore, pro-oncogenic reprogramming of the stroma is accompanied by an upregulation of ECM components, such as fibronectin [340]. Yalak et al. reported that CK2-mediated phosphorylation of fibronectin promoted cell spreading and metabolic activity in vitro [341].

GBM is characterized by diffuse and uncontrollable brain invasion. Recently, it has been shown that interferon regulatory factor 3 (IRF3), a transcriptional repressor of ECM factors, acts as a suppressor of GBM invasion. IRF3 has been previously determined to be negatively regulated by CK2 [342]. CK2 inhibition via TBB and CX-4945 therapeutically activated IRF3, downregulated the expression of ECM factors, and suppressed GBM invasion in vitro as well as in vivo [343].

Ku et al. demonstrated that CK2 inhibition by CX-4945 suppressed migration and invasion of the human lung cancer cell line A549 while simultaneously inhibiting the expression of membrane type 1-matrix metalloproteinase led to the selective attenuation of MMP-2, which can degrade components of the ECM [344]. These effects were attributed to the sequential attenuation of proteins in PI3K/AKT and MAPK pathways [344].

#### 2.4.6. Unbalanced Expression of CK2 Subunits

Unbalanced expression of CK2α and CK2β subunits has been described for many different mammalian tissues [345,346,347]. Furthermore, different substrate specificity has been detected for the CK2α and the CK2α’ subunits as well as the CK2 holoenzyme [348], leading to the definition of class I-III CK2 substrates [11]. Therefore, it becomes clear that the unequal expression of CK2 components has an important impact on the overall CK2 activity. In the context of invasion and metastasis the discrepancy in the expression of CK2 kinase subunits was described to drive EMT by Snail1 induction [270].

Snail1 is a zinc-finger transcription factor, which usually acts as a transcriptional repressor and has a pivotal role in the regulation of EMT [349]. Nuclear forkhead box protein C2 (FOXC2) is a transcriptional regulator of mesenchymal transformation during developmental EMT and has been associated with EMT in malignant epithelia. FOXC2 is functionally maintained in the cytoplasm by CK2α/α′-mediated phosphorylation at Ser124, promoting a normal epithelial phenotype [350]. The regulation of Snail1, as well as FOXC2, is dependent on the CK2β regulatory subunit. In epithelial cells, Snail1 stability is negatively regulated by CK2 and GSK3β, CK2β-depleted epithelial cells displayed EMT-like morphological changes, enhanced migration, and anchorage-independent growth, all of which require Snail1 induction [270]. In malignant breast cancer cells, the CK2β regulatory subunit is downregulated and the expression of FOXC2 in the nucleus correlated with the expression of mesenchymal genes [350]. This downregulation is sufficient to drive the early steps in tumor cell dissemination through the coordinated regulation of Snail1 and FOXC2 [270,350,351]. This demonstrates that not only CK2-mediated post-translational modifications but also the expression levels of the CK2 subunits significantly affect the process of invasion and metastasis. Since CK2α and CK2α’ maintain their kinase activity outside of their association with CK2β [2] an unbalanced expression of CK2 subunits might not influence its kinase activity but could interfere with the interaction of the CK2 tetramer with other cellular proteins.

All CK2-dependent regulated proteins, which are described in this chapter and are involved in invasion and metastasis are summarized in Figure 4.

### 2.5. CK2 Favors Metabolic Rewiring

Another hallmark of cancer is an reprogrammed energy metabolism, which can trigger chronically altered cell proliferation [13]. The first description of this mechanism was given by Otto Warburg who showed that in the presence of oxygen cancer cells can reprogram their glucose metabolism by limiting their energy metabolism mainly to glycolysis. This state has been termed the “aerobic glycolysis/Warburg effect” [352,353]. The enhanced growth and proliferation of cancer cells is supported by aberrant uptake and utilization of multiple nutrients, including glucose, glutamine, nucleotides, and lipids [354]. It has been previously described that CK2 is involved at many points in the malignant alteration of the energy metabolism of cancer cells thereby enhancing resistance to cell death caused by a microenvironment deficient in oxygen or nutrients (reviewed in [355]).

#### 2.5.1. Warburg Effect

The involvement of CK2 in the Warburg effect has been observed in CRC, esophageal, and bladder cancer cells [356,357]. In brief, Im et al. provided evidence for CK2-dependent, aerobic glycolysis-induced, elevated levels of lactate dehydrogenase A (LDHA), an enzyme that facilitates the glycolytic process by converting pyruvate to lactate [356]. The presence of increased LDHA in several cancer entities was also pointed out by Feng et al. who discussed LDHA as a potential target for cancer therapy [358]. Yang et al. could show that aerobic glycolysis was enhanced in stable cell lines expressing the CK2 catalytic subunit α accompanied by a downregulation of the pyruvate kinase M1, nuclear localization of pyruvate kinase M2 (PKM2) and a consequent elevation of LDHA expression and activity [357].

Implicated in the Warburg effect is the PI3K/AKT signaling network, which affects the cellular metabolism either by direct regulation of metabolic enzymes and nutrient transporters or via the control of transcription factors of components of metabolic pathways [359,360]. CK2 can interfere with this pathway e.g., by phosphorylating AKT on Ser129, which promotes AKT kinase activity [73]. AKT in turn post-translationally regulates metabolic enzymes such as thioredoxin-interacting protein (TXNIP), hexokinase 2 (HK2), 6-phosphofructo-2-kinase/fructose-2,6-biphosphatase (PFKFB2), transketolase (TKT) and NAD kinase (NADK), all of which contribute to anabolic metabolism [360]. The phosphorylation of AKT by CK2 is necessary for the AKT-dependent up-regulation of β-catenin transcriptional activity, a key protein in the canonical Wnt signaling pathway, which drives metabolic changes supportive of cancer metabolism [361,362].

In addition, CK2 is able to phosphorylate β-catenin directly in its armadillo repeat at Thr393, which protects β-catenin from proteasomal degradation and thereby promotes protein and co-transcriptional activity [24,25,26,27,363]. Furthermore, the phosphorylation of nuclear Lymphoid enhancer-binding factor 1 (LEF1) via CK2 significantly enhances the affinity of LEF1 for β-catenin and stimulates transactivation of the β-catenin/LEF1 complex thereby fostering initiation of the transcription of the target genes [28]. The interaction of CK2 with AKT and its stimulation of β-catenin further promotes the β-catenin-dependent expression of Survivin and enhances cell survival [364]. Survivin is upregulated in a variety of human cancers [140,141]. It has been shown to be localized to the mitochondria of certain tumor cell lines which enhanced the stability of oxidative phosphorylation Complex II, promoting cellular respiration [365]. Moreover, Survivin was demonstrated to support the subcellular trafficking of mitochondria to the cortical cytoskeleton of tumor cells. This was associated with increased membrane ruffling, increased focal adhesion complex turnover, and increased tumor cell migration and invasion in vitro. This effect also enhanced metastatic dissemination in vivo, therefore, contributing to an enhanced cancer metabolism [365].

Due to the dysfunctional vascularization, large areas of cancer tissue can develop a hypoxic environment, which promotes hypoxia-induced metabolic modifications and correlate metabolic change with hypoxic-adapted malignancy [366]. CK2 seems to be involved in this process by regulating *HIF-1α* [259], a protein that regulates the expression of secreted factors and supports tumor cell survival under hypoxic conditions [260,261,262]. However, although Mottet et al. showed in silico that HIF-1α contains five putative CK2 phosphorylation sites, Ser551, Ser581, Ser786, Thr700, and Thr796, they were not able to identify a direct CK2-dependent HIF-1α phosphorylation in their HepG2 cell model [259]. In contrast, the regulation of HIF-1α transcriptional activity seems to happen through the downregulation of VHL, a master regulator of HIF [264,265]. Ampofo et al. reported that CK2 phosphorylates VHL at Ser33, Ser38, and Ser43. Inhibition of CK2-mediated phosphorylation of VHL by TBB led to enhanced stability of the VHL protein and contributed to the degradation of HIF-1α [265]. Moreover, the aberrant activation of the Wnt/β-catenin pathway by CK2 and its intimate link with driving metabolic alterations of glycolysis, glutaminolysis, and lipogenesis [362] was shown to promote HIF-1α activity in a hypoxia-independent manner [367].

#### 2.5.2. Mitochondrial Metabolism

The metabolism of mitochondria is another very essential cellular process, which can be reprogrammed in cancer cells. Recently, mitochondrial metabolism has been shown to support tumor anabolism by providing key metabolites for macromolecule synthesis and generating oncometabolites to maintain the cancer phenotype [368,369] and CK2 has been demonstrated to be important in inhibiting mitochondria-mediated apoptosis in cancer (reviewed in [370]). The mitochondrial electron transport chain (ETC), nucleotide metabolism linked to mitochondrial ETC as well as the mitochondrial citric acid cycle have evolved as valuable targets in cancer therapy, also in combination with other anti-cancer agents [368]. Mitochondrial fusion contributes to restoring mitochondrial function by enabling the mixing of the contents of partially damaged mitochondria as a form of complementation. CK2 has been shown to be essential for promoting mitochondrial fusion in a Wnt/β-catenin-dependent manner [369]. Investigations on intracellular calcium (Ca^2+^) homeostasis in prostate cancer cells further linked CK2 with mitochondrial metabolism and revealed that CK2-inhibition by TBB and CX-4945 led to a concomitant increase in Ca^2+^ in the ER and mitochondria thereby contributing to the induction of apoptosis [371].

Another important function of CK2 in the modulation of mitochondrial metabolism was demonstrated by Dixit et al. in glioma cells [372]. Here, CK2 downregulated the mitochondrial protein pyruvate dehydrogenase kinase isozyme 4 (PDK4) to sustain the elevated energy demands in glioma cells and thus regulated cell proliferation. Upon CK2-inhibition by TBB, PDK4 inhibited glucose uptake through 5′-AMP-activated protein kinase (AMPK) catalytic subunit alpha-1 activation and maintained glioma cells in a chronic energy-deprived state ultimately leading them towards apoptosis [372].

The deregulation of lipid and cholesterol-associated signal transduction is another important mechanism of cancer cells. The role of CK2 in aberrant lipid metabolism in cancer has been reviewed recently [373].

#### 2.5.3. Autophagy

Autophagy is an intracellular recycling process that maintains basal levels of metabolites and biosynthetic intermediates under starvation or other forms of stress. This orchestrates metabolic adaptation in cancer cells which either impedes or promotes cancer initiation and progression [374]. CK2 has been shown to either directly or indirectly influence autophagy by modulation of involved key transcription factors. Olsen et al. provided evidence that siRNA-mediated downregulation of CK2 induced autophagic cell death through modulation of the mTOR and MAPK signaling pathways in human GBM cells [375]. Su et al. discovered that inhibition of CK2 by CX-4945 induced autophagy-mediated cell death through dephosphorylation of acetyl-CoA carboxylase in squamous cell carcinoma of head and neck cancer [376]. In addition, CK2-inhibition with CX-4945 has been shown to induce autophagy-triggered apoptosis when used alone in rat and human chondrocytes [377]. This inhibitor also promoted methuosis-like cell death associated with catastrophic massive vacuolization of CRC cells [378]. CK2 was further implicated in the activation of the transcription factor nuclear factor erythroid 2-related factor 2 (Nrf2), which plays an important role in maintaining intracellular redox homeostasis, via autophagic degradation of Kelch-like ECH-associated protein 1 (Keap1) and activation of AMPK in human breast cancer cells MCF-7 and human colon cancer cells HCT116 [379]. Activation of Nrf2 by CK2 is of importance since Nrf2 target genes, such as glutathione S-transferase, glutathione peroxidase 2, and glutathione reductase 1 contribute to the modifications within cancer metabolism as well as adhesion and metastasis [380,381,382].

As mentioned at the beginning of this chapter, nutrient deprivation is one of the hallmark conditions of the tumor microenvironment (TME) [383]. Fernandez-Saiz et al. could show that CK2 phosphorylates Ser485 of telomere maintenance 2 (Tel2) and Ser828 of Tel2 interacting protein 1 (Tti1) upon nutrient withdrawal [384]. This enables the E3 ubiquitin ligase SCFFbxo9 to target Tel2 and Tti1, which leads to the disassembly of the PI3K-related kinase complex mTORC1. Consequently, the mTORC1s inhibitory effect on mammalian target of rapamycin complex 2 (mTORC2) is abrogated which allows the cell to maintain its survival state even during nutrient deprivation [384].

Moreover, Park et al. demonstrated that CK2 acted as a positive regulator in calorie restriction and autophagy in nematodes [385]. Townley et al. provided evidence that Survivin, which has already been described in this review to be influenced by CK2 [146,364,365], is partially localized in mitochondria. Therefore, Townley and coworkers suggested that mitochondrial Survivin steers cells towards the implementation of the Warburg transition by inhibiting mitochondrial turnover, which enables them to adapt and survive [386]. CK2 has further been shown to link extracellular growth factor signaling to the control of p27 stability in the heart [387], a factor that positively regulates cardiac autophagy at rest and after metabolic stress in vitro and in vivo [388].

Taken together, it becomes clear that CK2 is involved in the Warburg effect, fosters changes within different signaling pathways as well as the mitochondrial and lipid metabolism and drives metabolic alterations, which are favorable for cancer cells.

All CK2-dependent regulated proteins, which are described in this chapter and which favor metabolic rewiring, are shown in Figure 5.

### 2.6. CK2 Abets the Tumor Microenvironment

The tumor microenvironment (TME) becomes formed during the course of multistep tumorigenesis and is the result of significant molecular, cellular and physical changes within the host tissues. Even though its distinct composition varies between different tumors the TME supports the survival, invasion, and metastatic dissemination of cancer cells [389]. CK2 has multiple roles in angiogenesis, invasion, metastasis as well as immune modulation (as described in this manuscript), all of which are important processes within the TME. Additionally, it has been shown that CK2 exerts an influence on the TME in other ways and thus supports cancer progression.

#### 2.6.1. Modulation of the Cytoskeleton, Microtubules, and Ion Channels

Changes in cytoskeletal protein association and regulation are fostered and exploited by cancer cells to facilitate metastasis (reviewed in [390]). CK2 has been shown to influence several compartments of the cytoskeleton e.g., microfilaments, microtubules, intermediate filaments, and septins, all of which differ concerning their mechanical stiffness, dynamic of assembly and the type of associated proteins within the cytoskeleton (reviewed in [391]). Usually, tumor tissues become denser than healthy tissues with more aligned fibers and changed porosity [392]. Wang et al. provided evidence that CK2 contributes to cytoskeletal reorganization [393]. They demonstrated that IR-induced cellular senescence of human mesenchymal stem cells (hMSC), which are critical for tissue regeneration. IR also induced dramatic cytoskeletal reorganization of hMSC, partially through redistribution of myosin-9. Using a SILAC-based phosphoproteomics method, they detected a significant reduction of myosin-9 phosphorylation at Ser1943, which coincided with its redistribution [393]. CK2 phosphorylates myosin-9 at Ser1943, and regulates the assembly and interactions of myosin-9 [394].

During actin polymerization in non-muscle cells, actin monomers are added to the barbed (fast-growing) ends of the filament, which are eventually capped by the actin capping protein to stop the growth and limit polymerization [395]. Canton et al. assumed that CK2 is involved in the regulation of the actin cytoskeleton [395]. They demonstrated that the Pleckstrin Homology Domain-Containing Protein Casein Kinase 2-interacting Protein 1 (CKIP-1) interacts with CK2 and actin capping protein subunits, CPα and CPβ. Furthermore, they showed that CPα is phosphorylated by CK2 at amino acid Ser9 in vitro and that the treatment with the CK2-inhibitor TBB resulted in a decrease in CPα phosphorylation. They concluded that both CKIP-1 and CK2 play a role in regulating cell morphology and the actin cytoskeleton [395]. If this phosphorylation is mediated by CK2 in vivo remains to be investigated.

Moreover, CK2 has been shown to phosphorylate HS1 in the N-terminal region and at different serine positions in the central core of the molecule [132]. In vitro experiments demonstrated that HS1, once phosphorylated by CK2, becomes resistant to cleavage by caspase-3 [133]. HS1 is known to promote actin polymerization by activating the Arp2/3 complex, and participates in multiple processes that remodel the actin cytoskeleton, including immunological synapse formation, cell adhesion, and migration [134].

Cilia are microtubule-based organelles, which are extended from the plasma membrane, coupled to the cytoskeleton, and are essential for the development and maintenance of adult tissues [396]. The Serine/Threonine kinase Tau tubulin kinase 2 (TTBK2), which contributes to the stability of the ciliary axoneme [397], has been shown to be regulated by CK2 [396]. Loukil et al. suggested, based on in vitro and in vivo analyses, that CK2 acts as a negative regulator of TTBK2 function, modulates the ciliary actin cytoskeleton, mediates ciliary trafficking, and ciliary tip stability, and prevents aberrant F-actin polymerization [396]. Changes in ciliation of cancer cells and/or cells of the TME during tumor development enforce asymmetric intercellular signaling in the TME [398]. Since CK2 is important in the regulation of cilia stability and is frequently overexpressed in cancer, one might speculate that this is an additional mechanism by which CK2 influences the TME.

The proteoglycan NG2 crucially determines the migratory capacity of distinct cancer cells and has been implicated in the tumorigenesis of colorectal cancer, glioma, and hepatocellular carcinoma [399,400,401]. CK2 has been reported in NG2 expression in juvenile angiofibroma (JA) [402]. Boewe et al. provided evidence that CK2-inhibition with CX-4945 and SGC-CK2-1 significantly reduced NG2 gene and protein expression. They demonstrated that this inhibition suppressed the proliferation and migration of JA cells [402]. Furthermore, Cattaruzza et al. detected that the binding of collagen type VI, an ECM ligand of NG2, to NG2, triggered the activation of cell survival- and cell adhesion/migration-promoting pathways and thereby influenced cancer cells and TME [403]. Moreover, CK2 has been shown to regulate NG2-mediated angiogenic activity of human pericytes [244]. Pharmacological inhibition of CK2 by CX-4945 suppressed pericyte proliferation, migration, spheroid sprouting, and the stabilization of endothelial tubes. In vivo, implanted Matrigel plugs containing CX-4945-treated pericytes exhibited a lower microvessel density than controls [244]. Taken together, the CK2-NG2 axis might be another signaling pathway to influence the TME.

In addition, CK2 has been shown to phosphorylate histone deacetylase 6 (HDAC6) at Ser458 thereby upregulating its deacetylase activity [404]. The relevance of this connection will be further discussed in the present review with regard to its MMR function [405,406]. HDAC6 has been reported to play a role in the dissemination and invasiveness of breast cancer [407], two processes that are promoted by the TME [408]. Recently, HDAC6 has been shown to induce the reorganization of the vimentin intermediate filament network, leading to increased cellular stiffness [409]. Although oncogenes simian virus 40 large T antigen, c-Myc, and cyclin E induced the up-regulation of HDAC6 and the altered spatial distribution of acetylated microtubules, the CK2-mediated upregulation of HDAC6 might still be relevant in this regard.

Microtubules control cell architecture by serving as a scaffold for intracellular transport, signaling as well as organelle positioning, and the remodeling of this network is implicated in changes within the TME [410]. CK2 seems to be involved in the regulation of microtubule reorganization. CK2α and CK2α’ have been shown to bind to tubulin and to modulate microtubule dynamics and stabilization [411]. As already mentioned, CK2 is able to phosphorylate HDAC6, which is also important concerning microtubules [404]. Furthermore, it has been reported that subcellular localization of the microtubule organizing center (MTOC) is modulated by ECM stiffness which helps to drive cell polarization during migration [412]. In addition, a role for CK2 in soft ECM has recently been demonstrated [413]. CK2 phosphorylated Ser236 in the C-terminus of Alpha-tubulin N-acetyltransferase 1 (α-TAT1) which was essential for its acetyltransferase activity. The acetylation of microtubule is an integral part of TGFβ-induced myofibroblast differentiation in the soft ECM. This suggests that CK2-mediated phosphorylation of Ser236 of α-TAT1 promotes microtubule acetylation upon TGFβ stimulation [413]. An increase in the level of microtubule acetylation is required for breast cancer progression [414] and is involved in forming an active TME through the modulation of myofibroblast differentiation [415]. Thus, CK2 may have a TME-promoting role through its influence on microtubules in soft ECM as well as stiff ECM.

The role of CK2 in the metabolic rewiring of cancer cells has already been discussed earlier. Another key feature of cancer metabolism is an increased acid production, which results in enhanced expression and/or activity of acid-extruding ion transport proteins [416]. Since CK2 is involved in cancer metabolism and has a significant impact on ion channels in the plasma membrane [417], one might speculate that CK2 also exerts an impact on acid-extruding ion transport proteins. The Ca^2+^-activated chloride channel TMEM16A is intimately associated with cancer and influences a variety of cancer hallmarks [418]. Using a microscopy-based high-throughput screening Pinto and coworkers demonstrated that CK2 is required for proper membrane expression of TMEM16A [419]. siRNA knockdown of CK2 as well as CK2 inhibition with TBB or CX-4945 reduced plasma membrane expression of TMEM16A and Ca^2+^-activated movements of ions across the cell membrane in cystic fibrosis bronchial epithelial airway cells as well as in the head and neck cancer cell lines Cal33 and BHY. This also had an effect on cell proliferation [419]. TMEM16A has been reported to interact with tyrosine 3-monooxygenase/tryptophan 5-monooxygenase activation protein gamma (14-3-3γ) which is suggested to influence the trafficking of TMEM16A to the plasma membrane [418]. 14-3-3γ is known to be phosphorylated by CK2 at Ser235 in vitro even though the in vivo consequences of this interaction remain to be elucidated [420]. Additionally, TMEM16A has been demonstrated to regulate IGF/IGFR signaling in the tumor microenvironment via the antiangiogenic factor IGFBP5 [421]. Since CK2 is implicated in the regulation of TMEM16A subcellular localization one might speculate that this interaction contributes to processes within the TME.

#### 2.6.2. Influence on Components of the Extracellular Matrix

Previously, CK2 has been demonstrated to function as an ectokinase, a kinase that phosphorylates extracellular proteins. While CK2′s export has been shown to be dependent on the CK2β subunit, the CK2 holoenzyme is required for its ectokinase function [422,423]. ECM is an important feature of the TME and ECM destruction is associated with cancer cell invasion [339,424]. Yalak et al. elucidated the role of CK2-mediated phosphorylation of ECM proteins. They demonstrated in vitro by using nanopillar arrays that CK2-dependent phosphorylation of fibronectin significantly upregulated cell traction forces and total strain energy, as well as cell spreading and metabolic activity [341]. Fibronectin is a key component of the ECM and is highly upregulated in cancer [425]. Published proteomic data demonstrated that fibronectin is heavily phosphorylated in domains responsible for growth factor signaling and fibrillogenesis in clinical cancer tissue samples [341,426].

Recently, it has been shown that IRF3, a transcriptional repressor of ECM factors, acts as a suppressor of GBM invasion. IRF3 has been previously determined to be negatively regulated by CK2 [342]. CK2 inhibition via TBB and CX-4945 therapeutically activated IRF3, downregulated the expression of ECM factors, and suppressed GBM invasion in vitro as well as in vivo [343].

MMPs are responsible for the degradation of ECM proteins such as collagen and enable tissue remodeling and repair [427]. Ku et al. demonstrated that CK2 inhibition by CX-4945 suppressed migration and invasion of the human lung cancer cell line A549 and simultaneously inhibited the expression of membrane type 1-matrix metalloproteinase (MMP-1), and led to the selective attenuation of MMP-2. These effects were attributed to the sequential attenuation of proteins in the PI3K/AKT and MAPK pathways [344]. Furthermore, CK2 has been shown to phosphorylate and inhibit MMP-2 in vitro, an effect that could be abolished by the inhibition of CK2 with TBB [428].

Integrins, a diverse family of cell attachment receptors, are commonly dysregulated in many cancers and play a major role in signaling, promotion of tumor cell invasion, and adhesion of circulating tumor cells [429]. Zheng et al. provided evidence that CK2 is also implicated in the regulation of integrins [57]. Inhibition of CK2 activity by CX-4945 or CK2 knockdown by siRNA suppressed the activation of the JAK/STAT, NF-κ,B and AKT pathways. This has been mediated in part through inhibition of integrin α4 and β1 expression [57]. Integrin α4 has been implicated in drug resistance of leukemia [430] while integrin β1 has been reported of relevance in invasive breast cancer [431]. Moreover, the α4β1 integrin heterodimer plays a crucial role in inflammation, the interaction with its ligand Vascular cell adhesion protein 1 increases transendothelial migration and contributes to metastasis [432].

Vn, an adhesive glycoprotein linking cells and the ECM through ligands such as integrins, is associated with tumor invasion, metastasis, and angiogenesis [433]. Seger et al. demonstrated that CK2 can phosphorylate Vn at Thr50 and Thr57 and enhance cell adhesion and spreading of bovine aorta endothelial cells [281]. They further elucidated the mechanism of this phosphorylation and found that this phosphorylation increased cell adhesion via the integrin α5β3 heterodimer. The enhanced cell adhesion coincided with a preferential activation of the FAK/PI3K/AKT cascade, rather than the ERK signaling pathway, thereby implicating CK2-mediated phosphorylation in the activation of distinct signaling cascades [282]. Further investigations concerning the interplay of integrins and CK2 are appropriate, since integrins are present in the tumor microenvironment of all cell types and play important roles in the regulation of intercellular communication [434].

#### 2.6.3. Cancer Stem Cells

Cancer stem cells (CSCs), a small subpopulation of cells within tumors with capabilities of self-renewal, differentiation, and tumorigenicity, regulate multiple cancer hallmarks through their interaction with cells and ECM in their environment (reviewed in [435]). CSC phenotype is dependent on the TME and CK2 has already been linked to CSC in various studies. Lu et al. reported that phosphorylation of TAp73, a structural homologue of p53, by CK2 inactivates this tumor suppressor and promotes the expression of CSC genes and the CSC phenotype [203,204]. Furthermore, it has been suggested that CK2 may interfere with human stem cell recruitment during angiogenesis. Kramerov and coworkers demonstrated that bone marrow-derived human stem cells injected into the vitreous of neonatal mice could incorporate into the retinal neovasculature, a process that has been significantly reduced by CK2 inhibitors emodin, DRB, TBBt, and TBBz [247].

The Hh signaling pathway plays important roles in mammalian development and in stem cell maintenance [436,437]. CK2 has been shown to increase Hh signaling in lung cancer cell lines A549 and H1299 [333]. The inhibition of Hh by CK2α siRNA reduced CSC-like side populations [333].

Furthermore, CK2 has been reported as a positive regulator of Notch1 signaling in lung cancer cell lines A549 and H1299 [311]. Notch1 has been implicated in stem cell maintenance. CK2 inhibition with CX-4945 led to a dose-dependent inhibition of Notch1 transcriptional activity while the overexpression of CK2α increased Notch1 transcriptional activity. Moreover, the inhibition of CK2α led to a reduced proportion of stem-like CD44+/CD24− cell population, thereby providing further evidence for the influence of CK2 on CSC [311].

Basu et al. reviewed the connections between the Wnt/β-catenin signaling pathway, EMT, and cancer cell stemness [438]. An important factor herein was Zeb1, a transcription factor that inhibits E-cadherin transcription. An increased expression of Zeb1 has been reported in tumor-initiating stem-like cells in colorectal and pancreatic cancer [439]. Deshiere et al. provided evidence that CK2β subunit silencing triggers EMT-like morphological changes in the breast cancer cell line MCF10A with a simultaneous upregulation of Zeb1 [270]. Chung et al. performed in silico protein analysis and identified a putative CK2 phosphorylation site at residues Ser1036 to Glu1039 of Zeb1. They suggested that the loss of this putative CK2 phosphorylation site causes inactivity, and subsequent haploinsufficiency of Zeb1 in posterior polymorphous corneal dystrophy 3 [440].

The ET-1 axis contributes to the pathophysiology of several cancers by promoting tumor development and progression [323]. Different groups could show that CK2 phosphorylates Ser18 and Ser20 in the N-terminus of the isoform of Zinc metalloprotease ECE-1c, which enhances the stability of ECE-1c, and promotes stem cell traits, invasiveness, and aggressiveness of CRC cells [324,325,326].

Liu and coworkers detected that Emodin, which inhibits CK2, suppressed EMT and CSC formation of breast cancer cells [441]. This effect has been mediated by the blockage of TGFβ1-dependent crosstalk between tumor-associated macrophages (TAMs) and breast cancer cells [441]. Whether this mechanism is exclusively dependent on TGFβ1 or might be mediated by distinct actions of CK2 remains to be elucidated. Recently, Zheng et al. reviewed how the release of cytokines and exosomes within the TME is involved in inducing non-CSCs to acquire CSC properties and increasing CSC plasticity [442]. Since CK2 is also implicated in the regulation of cytokine signaling [62], this might be another mechanism by which it influences the TME. Furthermore, CK2 has been linked to triple-negative breast CSCs [443] and has been suggested in CSC maintenance in GBM [444].

#### 2.6.4. Modulation of the Immune Compartment in the TME

Myeloid cells are the most abundant cells in the TME and are kept at an immature state of differentiation to be diverted to an immunosuppressive phenotype in cancer, thereby sustaining a beneficial environment for tumorigenesis [445]. CK2 has been implicated in the differentiation of myeloid cells by various groups. Zheng et al. provided evidence that increased activity of CK2 is responsible for Notch phosphorylation and downregulation in hematopoietic progenitor/stem cells (HPC) and myeloid-derived suppressor cells (MDSC) [446]. CK2-inhibition with siRNA and pharmacological inhibition with TBCA restored the Notch signaling in myeloid cells and significantly improved their differentiation [446]. In addition, Hashimoto et al. reported that the inhibition of CK2 with BMS-211, a prodrug of the parent pan-CK2 inhibitor BMS-699, substantially reduced the amount of polymorphonuclear myeloid-derived suppressor cells (PMN-MDSC) by blocking the differentiation in mice [447]. They suggested that the modulatory effects of CK2 inhibitors on myeloid cell differentiation in the TME might be synergized with the anti-tumor effects of immune checkpoint receptor blockade [447].

Fibroblasts can differentiate from myofibroblasts, which orchestrate tissue repair and wound healing, mediated by ECM remodeling and their crosstalk with innate immune cells. In the TME the physiological functions of fibroblasts are dysregulated [448]. Different groups have implicated CK2 interaction with fibroblasts and fibroblast growth factors. Zhang et al. provided evidence that CK2 is activated in Systemic sclerosis (SSc), a chronic fibrotic disease [449]. CK2 activation contributed to fibroblast activation by regulating the JAK2/STAT3 signaling pathway. Inhibition of CK2 with TBB abrogated the pro-fibrotic effects of TGFβ and inhibited experimental fibrosis [449]. Using U2OS human osteosarcoma cells Skjerpen et al. demonstrated an interaction between Fibroblast growth factor-1 (FGF-1) and CK2 [450]. The group showed in vitro that CK2 phosphorylated FGF-1 and FGF-2 and reported that FGF-1 and CK2 are able to interact in vivo [450]. Bonnet and coworkers presented that FGF-2 directly interacts with CK2, thereby stimulating CK2 activity toward nucleolin [451]. Nucleolin phosphorylation by CK2 has been shown to control cellular fate by regulating the p53 checkpoint under normal and stressed conditions [452]. Furthermore, Nucleolin has been implicated in stemness and carcinogenesis and has been detected in the cell membrane of multiple cellular subpopulations of the TME, including CSC [453,454]. Moreover, CK2 has been reported to be involved in cytokine production, a key process ofIR-inducedduced perivascular resistant niche, by endothelial cells, which led to radioresistance of NSCLC cells [455].

Cyclooxygenase-2 (COX-2), which converts arachidonic acid into thromboxanes and prostaglandins, has been reported to be a key molecule mediating TME changes and contributes to immune evasion as well as resistance to cancer immunotherapy [456,457]. The inflammatory mediator prostaglandin E2 (PGE2) has been shown to play a central role in tumorigenesis [458]. Yefi et al. elucidated the connection between CK2 and COX-2 [459]. They provided evidence that COX-2 expression and cell viability decreased upon inhibition of CK2 with DMAT in human colon cell lines, HT29-ATCC, HT29-US, DLD-1, breast cancer cell line ZR-75 as well as HEK293T. The ectopic expression of CK2α activated the Wnt/β-catenin pathway in HEK293T cells, and promotes an up-regulation of COX-2. The overexpression of COX-2 as well as the supplementation of medium with PGE2 abrogated CK2-inhibitor-induced changes in cell viability [459].

Angiotensin-converting enzyme (ACE), the key peptidase of the renin angiotensin system (RAS), has been implicated in carcinogenesis, and has recently been recognized in modulating the immune compartment of the TME [460]. CK2 has been shown to phosphorylate ACE in the cytoplasmic tail at Ser1270 in endothelial cells [461]. Proteolytic cleavage of the C-terminal domain of ACE derives it from the membrane-bound form to the soluble form. Kohlstedt and coworkers provided evidence that dephosphorylated ACE is not retained in the plasma membrane [461]. Treatment of endothelial cells with CK2-inhibitor DRB decreased the phosphorylation of ACE and increased its shedding. Therefore, Kohlstedt et al. suggested that CK2-mediated phosphorylation of ACE regulates its retention in the plasma membrane and may determine plasma ACE levels [461]. This, together with the fact that ACE can function as a signal transduction molecule [462] and regulates cell proliferation and migration [463] might drive immune modulations within the TME.

The involvement of CK2 with HIF-1α has been described in “CK2 induces angiogenesis and vascularization” as well as “CK2 favors metabolic rewiring”. Since CK2, hypoxia, and HIF-1α are closely associated with the TME [464,465], it might be assumed that these CK2-mediated signaling cascades contribute to the maintenance of TME effects.

All CK2-dependent regulated proteins, which are described in this chapter and which are associated with the tumor microenvironment, are shown in Figure 6.

### 2.7. CK2 in Genome Instability and Mutation

DNA repair pathways maintain genetic stability when mammalian cells are exposed to endogenous or exogenous DNA-damaging agents [466]. Genomic instability and thus mutability is reported as an important driver of tumor progression and has been recognized as a hallmark of cancer [13,467]. CK2 has a variety of functions in DNA damage repair (DDR), it is in part essential for the functionality of the repair pathways, but it can also trigger genomic instability (reviewed in [8]). Several repair pathways exist, including direct reversal repair, base excision repair (BER), nucleotide excision repair (NER), mismatch repair (MMR), homologous recombination (HR), non-homologous end joining (NHEJ) as well as alternative end joining (alt-EJ) [468].

Direct reversal repair mainly repairs lesions induced by alkylating agents and contains, among others, O6-alkylguanine-DNA alkyltransferases (AGT) and AlkB family dioxygenases [469]. CK2 has been shown to phosphorylate AGT in the human medulloblastoma cell line UW228 [470]. This phosphorylation, in conjunction with phosphorylation of recombinant AGT by PKA and PKB, reduced AGT activity by 30–65% and thereby attenuated the repair [470]. AlkB homolog 5 RNA demethylase (ALKBH5), which belongs to the AlkB family of dioxygenases and reverses N-alkylated base adducts, is associated with CK2 as well. Yu et al. provided evidence that ALKBH5 inhibited the progression and sensitized bladder cancer cells to cisplatin which was dependent on N6-methyladenosine and mediated through a CK2α-dependent glycolysis pathway [471].

BER is a highly conserved mechanism dealing with oxidative damage generated by respiration, natural hydrolysis, and alkylation reactions which is often exploited by cancer cells to tolerate oxidative stress [472]. CK2 has been demonstrated to phosphorylate DNA repair protein X-ray repair cross-complementing protein 1 (XRCC1), a protein involved in the ligation step of both BER and repair of DNA single-strand breaks [473]. Ström et al. provided evidence in EM9 cells that CK2-mediated phosphorylation of XRCC1 attenuated its DNA binding ability [474]. Parsons and coworkers further elucidated this mechanism and reported that XRCC1 phosphorylation by CK2 is essential to stabilize a complex consisting of XRCC1 and DNA ligase III (Lig III), which is involved in the ligation step of both BER and repair of DNA single-strand breaks [473].

Another interaction that links CK2 and BER takes place via the chromatin transcriptional elongation factor FACT (FACT), a heterodimer of FACT complex subunit SPT16 (hSpt16) and Structure Specific Recognition Protein 1 (SSRP1) [475]. Upon oxidative stress, Richards et al. reported co-remodeling activity of FACT and strong enhancement of the remodeling capacity of ATP-dependent chromatin remodeler RSC [476]. This is thought to contribute to the excision of DNA lesions during the initial step of BER [476]. CK2α has been shown to associate with SSRP1, an association that increases in response to UV irradiation [475]. Li et al. reported that CK2 phosphorylates SSRP1 at Ser501 in vitro in response to UV but not gamma irradiation thereby regulating its DNA binding activity [477]. Moreover, Keller and coworkers identified CK2 after UV-irradiation in a complex with FACT, purified from F9 and HeLa cells, that phosphorylates Ser392 of p53 in vitro [478]. In this study, FACT has been shown to alter the specificity of CK2 in the complex leading to a selective phosphorylation of p53 over other substrates, and to enhance p53 activity. Further elucidation of this interaction by Keller et al. demonstrated that UV irradiation induced the association of the CK2, hSpt16, and SSRP1 complex, and increased the specificity of CK2 for p53 [479]. The Ser392 phosphorylation of p53 has been reported to influence its mitochondrial translocation and transcription-independent apoptotic function [480]. Taken together, this implicates CK2 in enhancing DDR upon UV irradiation, thereby contributing to the survival of cancer cells.

NER corrects bulky, helix-distorting DNA lesions [466]. Xeroderma pigmentosum group C (XPC) protein is implicated in NER [481] and has been shown to be phosphorylated by CK2 at Ser94 following UVB exposure. This fostered the recruitment of XPC and downstream NER factors to DNA damage sites, thereby promoting NER repair [482]. CK2 is further linked to XPC by its phosphorylation of centrins, which are interaction partners of XPC [481]. CK2 phosphorylates centrin 1 at Thr138 and centrin 2 at Thr138 and Ser158, reducing their binding to XPC [483]. However, the in vivo consequences of this phosphorylation for NER remain to be elucidated. Furthermore, CK2 has been demonstrated to phosphorylate two components of the transcription factor TFIIH complex, a core component of NER [484,485,486]. CK2 phosphorylated Xeroderma pigmentosum group B (XPB) in the C-terminus at Ser751 in vivo, which impaired NER but did not interfere with XPBs helicase activity [485]. CK2 also phosphorylated cyclin H at Thr315, which did not influence the assembly of the cyclin H/cdk7/Mat1 complex but turned out to be critical for a full cyclin H/cdk7/Mat1 kinase activity [486].

The MMR system repairs base-base mismatches and insertion/deletion loops (IDLs) [466] which arise either during DNA replication or are caused by DNA damage [487]. Christmann et al. provided evidence that CK2 phosphorylates the major mismatch-binding proteins MSH2 and MSH6, which form the MutSα complex, in vitro and in vivo. MSH6 was more extensively phosphorylated than MSH2 and the phosphorylation resulted in increased mismatch binding by MutSα [488]. Kaliyaperumal and coworkers suggested a model in which the degree of MSH6 phosphorylation and differential binding activity of MutSα are required to trigger specific cellular pathway responses [489]. Edelbrock et al. further elucidated the role of phosphorylation of MSH6 [490]. Mutation of a serine cluster containing only CK2 recognition motifs resulted in a dramatic decrease in binding affinity to G:T mismatches compared to O6meG:T whereas increased phosphorylation of MSH6 showed the opposite effects. This confirmed that the amount of MSH6 N-terminal phosphorylation (more for G:T than O6meG:T) and binding stoichiometry of MutSα to a specific lesion (higher for O6meG:T than G:T) contribute to the differentiation between MMR pathway signaling and MMR-dependent alkylation damage signaling [490].

Concerning the regulation of MSH2, OTUB1 has been hypothesized to be important. OTUB1 has been shown to inhibit MSH2 ubiquitination by blocking the E2 ligase ubiquitin transfer activity thereby stabilizing MSH2 [491]. Herhaus et al. demonstrated that CK2 phosphorylates OTUB1 at Ser16, which led to the localization of OTUB1 in the nucleus, and was essential to repairing IR-induced DNA damage in osteosarcoma U2OS cells [226]. This suggests that CK2 may control OTUB1 activity by restricting its access to substrates in the cytosol or nucleus and might contribute to the stabilization of MSH2.

Another possible mechanism of CK2 to influence the functionality of MSH2 seems to proceed via the phosphorylation of Brahma-related gene 1 (Brg1), a catalytic subunit of the mammalian SWI/SNF chromatin-remodeling enzymes [492]. Padilla-Benavides et al. provided in silico data of 10 potential phosphorylation sites for CK2 in Brg1 and in vitro data suggesting the C-terminal end of Brg1 as a dominant CK2 phosphorylation site. CK2 inhibition by TBB impaired Brg1 chromatin remodeling and transcriptional activity at the Pax7 locus. The CK2-mediated phosphorylation also correlated with the subunit composition of the SWI/SNF enzyme complex and its subnuclear localization [492]. The group further reported that the phosphorylation at the C-terminal end of Brg1 regulated Brg1s subcellular localization [493]. The SWI/SNF chromatin remodeler Brg1 has been recently shown to be essential for the chromatin remodeling function of MSH2 [494]. Nargund et al. demonstrated that MSH2 genomic binding in gastric cancer was associated specifically with tumor-associated super-enhancers controlling the expression of cell adhesion genes. This binding has been required for chromatin rewiring, was independent of MSH2′s DNA repair catalytic activity and the SWI/SNF chromatin remodeler Brg1 was a prerequisite for recruitment to gene loci [494]. However, whether the CK2-mediated phosphorylation of Brg1 is directly involved in this process remains to be elucidated.

Whereas phosphorylation by CK2 appears to contribute to an activation of the MutSα complex, our group in contrast found the decreased activity of the second most important MMR complex MutLα, consisting of MLH1 and PMS2, after CK2-dependent phosphorylation [495]. Using an in vitro kinase assay we found that MLH1 is a substrate of CK2α and our MMR repair assay demonstrated that phosphorylation of MLH1 at amino acid position Ser477 can switch off MMR activity in vitro. We hypothesized that this posttranslational modification might inhibit the ability of MLH1 to interact with MMR process essential proteins or might lead to allosteric conformational changes, inhibiting the endonuclease activity of MutLα [495], which is the current objective of our research.

Furthermore, we were able to verify the importance of CK2α-mediated MLH1 phosphorylation in vivo [496]. In a cohort of 165 CRC patients, we could show that enhanced MLH1 phosphorylation was significantly associated with high nuclear/cytoplasmic CK2α expression. In addition, high nuclear/cytoplasmic CK2α levels could be significantly correlated with a reduced 5-year survival outcome of patients. Furthermore, we detected enriched somatic mutation rates in tumors with high nuclear/cytoplasmic CK2α expression and speculated that enriched somatic mutation rates are induced by the reduction of MMR, caused by CK2-mediated phosphorylation of MLH1 [496]. Recently, MMR deficiency and high rates of microsatellite instability have been shown to be predictive of anti-programmed cell death 1 (PD-1)/PD-L1 immunotherapy efficacy. However, CRC durable responses with PD1/PD-L1 inhibitors can be achieved in only approximately 40% of patients with MMR-deficient tumors. Simultaneously, 3% of CRCs are MMR proficient, microsatellite stable and present a high tumor mutational burden which might classify them for immune checkpoint inhibitor-based therapeutic approaches [497,498,499,500]. Furthermore, it has been demonstrated that phosphorylation of PD-L1 at Thr285 and Thr290 by CK2 disrupted PD-L1 binding with speckle-type POZ protein and protected PD-L1 from proteasomal degradation in cancer [322]. Whether CRCs with CK2-dependent loss of MMR and related increase of mutation rates might be a target for anti-PD-L1 immunotherapy remains to be investigated.

Another possible signaling pathway, which implicates CK2 in the regulation of MMR, is the phosphorylation HDAC6. Watabe et al. provided evidence that CK2 phosphorylates HDAC6 at Ser458 which upregulated its deacetylase activity [404]. HDAC6 regulates DNA damage response and MMR activities via regulation of MutSα homeostasis through sequential deacetylation and ubiquitylation of MSH2. Additionally, HDAC6 has been demonstrated to significantly reduce cellular sensitivity to DNA-damaging agents and decrease cellular MMR activities by the downregulation of MSH2 [405]. Zhang et al. reported on a similar mechanism regarding MLH1. HDAC6 has been shown to interact with and deacetylate MLH1 both in vitro and in vivo, leading to the disruption of the assembly of the MutSα–MutLα complex [406]. Taken together, this implicates CK2 in the direct and indirect modification of MMR, leading to tolerance of DNA damage and thereby fostering cancer progression.

In addition, CK2 has been demonstrated to be involved in HR, one of two pathways essential to repairing DNA double-strand breaks (DSBs). HR is initiated by DSBs sensors such as H2A Histone Family, Member X (H2AX), and Meiotic recombination 11 homolog 1 (MRE11) [468]. Yang et al. could show in stable cell lines overexpressing CK2α that PKM2 was localized to the nucleus [357]. Recently, it has been reported that nuclear PKM2 interacts with H2AX under DNA damage conditions [501]. In vitro kinase assay revealed that PKM2 directly phosphorylated H2AX at Ser139 and the use of a kinase dead variant led to decreased cell proliferation and chromosomal aberrations under DNA damage conditions [501]. Therefore, it could be proposed that CK2-mediated nuclear localization of PKM2 influences genomic stability in tumor cells, which directly involves phosphorylation of H2AX and thereby indirectly the initiation of HR.

Moreover, CK2 has been implicated in HR via its influence on RNA polymerase III [502,503,504] and in the NHEJ pathway via its phosphorylation of DNA repair protein XRCC4 [505,506]. MRE11, ATP-binding cassette (ABC)-ATPase (RAD50), and Nijmegen breakage syndrome protein 1 (NBS1) form the MRN complex, which is essential in the sensing and repair of DSBs and is engaged in various DDR pathways [507]. CK2 has been shown to influence MRE11 stability [508] and to be involved in targeting NBS1 to DSBs [509].

The DNA repair protein apurinic endonuclease (APE/Ref-1) is a multifunctional protein and plays a central role in the classical NHEJ pathway [510]. Several years ago, Yacoub et al. demonstrated a CK2-mediated phosphorylation of APE/Ref-1 which abolished DNA repair activity in vitro [511]. In contrast, a thereupon following study by Fritz and Kaina showed that APE/Ref-1 phosphorylation by CK2 enhanced redox activation of the AP-1 transcription factor and had no effect on its DNA repair activity [512]. The discrepancy of both studies and the lack of knowledge about APE/Ref-1 phosphorylation sites in vivo requires further investigation to clarify the findings, especially since CK2 and APE/Ref-1 are both implicated in the regulation of DNA damage [510].

Palma et al. demonstrated that CK2 phosphorylates Ser87 of the Crossover junction endonuclease MUS81 (MUS81) [513]. The MUS81 complex resolves branched DNA intermediates in mitosis, crucially preserving genome stability. Phosphorylated MUS81 has been shown to interact with Structure-specific endonuclease subunit SLX4 (SLX4), thereby promoting the function of the MUS81 complex. The authors speculated that CK2-overexpression might elevate the biological function of MUS81 and thus contribute to genome instability [513].

Apart from its role in distinct DDR pathways, CK2 is involved in processes enabling DDR in general. For DDR, the respective proteins must access DNA damage sites, which is facilitated by heterochromatin relaxation. Ayoub et al. provided evidence that CK2 is involved in this process [514]. Heterochromatin is packed and maintained among other factors via heterochromatin protein 1 (HP1) binding to histone H3 lysine 9 trimethylation (H3K9me3). Upon DNA damage, CK2 phosphorylated HP1β at Thr51 and disrupted the HP1β interaction with H3K9me3, which induced transient heterochromatin relaxation [514]. A similar process is governed by SIRT6, which has been implicated in BER, DDR, and NHEJ [515,516,517]. SIRT6 has been shown to translocate to DNA damage sites, where it recruits and interacts with Chromodomain-helicase-DNA-binding protein 4 (CHD4), which in turn displaces HP1 from H3K9me3 and leads to chromatin relaxation [518]. Bae et al. characterized the relationship between CK2 and SIRT6 in breast carcinoma. They showed that CK2 is able to phosphorylate SIRT6 at Ser338 [216]. In addition, Hussein et al. provided in vitro and in vivo evidence that the CK2-mediated phosphorylation of SIRT6 was associated with the induction of the DDR pathway proteins [217]. The overexpression of CK2 could therefore accelerate DDR via these mechanisms and facilitate faster DDR in cancer cells.

Recently, the circadian clock has been linked to tumorigenesis and genome instability [517,519,520,521]. How circadian clock perturbation impacts genome stability thereby triggering tumor initiation and progression has been reviewed extensively [517]. CK2 is involved in this process e.g., via direct regulation of circadian proteins. CK2α has been shown to phosphorylate Brain and muscle Arnt-like protein-1 (BMAL1) at Ser90 [522]. BMAL1 is a central protein for the clock mechanism by binding to promoter E-box elements. The CK2-dependent phosphorylation of BMAL1 controls its nucleocytoplasmic localization and has been demonstrated to be an integral part of the core clock oscillator [522,523]. CK2 has also been reported to phosphorylate Period (Drosophila) Homolog 2 (PER2) at Ser53 which influenced circadian rhythms and regulated PER2 stability [524]. Another substrate of CK2, SIRT1, is required for high-magnitude circadian transcription of several core clock genes, including BMAL1 and PER2 [525]. Moreover, SIRT1 has been described as an important player in the DNA damage response [526]. SIRT1 functions as a histone deacetylase at DNA damage sites, and alters the condensation of the chromatin but also deacetylates proteins involved in DNA repair and DNA damage response [527]. After IR-induced DNA damage, CK2 has been reported to phosphorylate Ser659 and Ser661 of SIRT1 in vitro and Ser154, Ser649, Ser651, and Ser689 of SIRT1 in vivo [106,211]. Latter phosphorylation increased SIRT1s deacetylation rate and increased its substrate-binding affinity [209,212]. Since CK2 and SIRT1 are both implicated at various points in the regulation of the circadian clock as well as DNA repair mechanisms and also interact with one another this could contribute to influencing genome stability [517]. Moreover, CK2 is involved in gene control in cell cycle entry [528] and influences proteins in DNA replication and replication checkpoint control [529]. Therefore, CK2 overexpression might contribute to the alteration of the duration of cells at different phases of the cell cycle which has been shown to compromise genome stability [530,531].

Chemotherapy and radiotherapy of cancers aim to kill cancer cells primarily by causing DNA damage [532,533]. However, a deregulation of DNA repair pathways is often present in tumors and associated with cancer development and progression. Thus, specific targeting of DNA repair pathways can increase the tumor sensitivity to these therapies [534]. In this regard, many research groups consider CK2 to be a promising target. Wang et al. reported the CK2 inhibitor HY1 which, once conjugated with an active Pt(II) unit, reversed cisplatin-induced resistance in human lung cancer cell line A549 in vitro. This treatment also presented acceptable pharmacokinetic behavior and exhibited higher tumor growth inhibitory efficacy than cisplatin either in A549 or A549/cisplatin xenograft models with low toxicity [535]. Zhang et al. provided evidence for a novel dual-target inhibitor of Bromodomain-Containing Protein 4 (BRD4)/CK2, 44e [536]. BRD4 is a chromatin reader protein that recognizes and binds acetylated histones with a key role in the transmission of epigenetic memory and transcription regulation. Zhang and coworkers could show in vitro that 44e inhibited proliferation and induced apoptosis and autophagy-associated cell death of human breast adenocarcinoma cell lines MDA-MB-231 and MDA-MB-468. 44e displayed compelling anticancer activity without obvious toxicities in vivo [536].

Giacosa et al. provided evidence that combined inhibition of CK2 and ATM kinases with CX-4945 and KU-60019 in renal tumor cells and patient-derived tumor samples induced synthetic lethality. Mechanistically, this has been conveyed through ROS overproduction leading to apoptosis [537]. Kildey et al. could show that CK2 modulated the degradation of cell division cycle associated protein-3 (CDCA3) through the promotion of the interaction between CDCA3 and cofactor Cdh1 [538]. They demonstrated in vitro and in patients that CDCA3 levels correlated with genome instability and platinum sensitivity and reported that CDCA3 high tumors were sensitive to cisplatin and carboplatin. Inhibition of CK2 with CX-4945 disrupted CDCA3 degradation, which led to elevated CDCA3 levels and increased sensitivity to platinum agents. Moreover, Kildey and coworkers suggested that this mechanism could be exploited for a new form of combinatorial treatment in NSCLC [538]. Furthermore, Nitta et al. reported that CK2-inhibition by CX-4945 increased the sensitivity of medulloblastoma to temozolomide (TMZ), an orally administered alkylating agent. The synergistic effect was based on the reduction of β-catenin by CK2-inhibition, which in turn reduced O-6-methylguanine-DNA methyltransferase (MGMT), thereby increasing the sensitivity of medulloblastoma cells to TMZ treatment [539].

Yu suggested CK2 as a potential PARP regulator [540]. It has been shown in vitro that CK2 is able to interact with PARP1 in the nucleus and can phosphorylate it. In vitro and in vivo analyses showed that the combination of CK2 and PARP1 inhibition synergistically attenuates DDR, cell cycle, cell proliferation and xenograft tumor growth. Therefore, it has been implicated that the combined inhibition of CK2 and PARP1 might be a new therapeutic option for triple-negative breast cancer patients resistant to PARP-inhibition [540]. Taken together, combined inhibition of CK2 and distinct regulators of the DNA damage response might provide beneficial therapy outcomes.

All CK2-dependent regulated proteins, which are described in this chapter and are involved in genome instability are summarized in Figure 7.

### 2.8. CK2 Contributes to the Avoidance of Immune Destruction

The main function of the mammalian immune system is to protect against invading or infectious pathogens and to eliminate damaged cells while monitoring tissue homeostasis. The relationship between immune cells and developing tumors is complex as cancer cells employ various mechanisms mimicking peripheral immune tolerance thereby avoiding detection and elimination [541]. Recent studies showed that CK2 is an important factor for the normal physiological function of immune cells but also of significant relevance in the modulation of inflammation [6]; therefore, it is reasonable to assume that CK2′s involvement with the immune system also contributes to tumorigenesis. However, the roles of CK2 in the regulation of immune cells are incompletely understood so far.

#### 2.8.1. CK2 and the Innate Immune System

Innate immune cells are able to induce inflammation in situ thereby promoting tumor progression and orchestration of the activity of adaptive immune cells [542,543]. To elucidate the regulatory role of CK2 on innate immune cells, a conditional deletion Csnk2 floxed Lyz2-Cre mouse model was created by Larson et al., in which the catalytic CK2α subunit is deleted in myeloid cells [544]. Using systemic *Listeria monocytogenes* infection, it could be demonstrated that loss of CK2α did not affect the development of myeloid cells but enhanced their recruitment and anti-bacterial activity. Specifying the impact of CK2α on bactericidal potential and activation of myeloid cells Larson and coworkers further detected that Class II MHC expression as well as phagosome maturation was enhanced in case of CK2α deficiency. They suggested that CK2α attenuates early myeloid cell activation and accumulation at sites of infection [544].

In addition, several groups used in vitro experiments to examine the role of CK2 in innate immunity. Here, a critical role for CK2 was identified in monocytes on Interferon Regulatory Factor 4 (IRF4) NF-κB-regulated transcription of the cytokine IL-1β [545]. The IL-1β promoter, packaged into a non-transcribed “poised” configuration, is associated with two transcription factors, PU.1 and C/EBPβ. Upon LPS stimulation, IRF-4, a third transcription factor, can be recruited to the IL-1β promoter. For IRF-4 assembly the phosphorylation of PU.1 at Ser148 by CK2 is essential which allows the transcription and expression of IL-1β in monocytes [546,547]. In another study, it has been speculated that the inhibition of CK2, JNK, AP-1 and NF-κB as well as the inhibition of pro-inflammatory mediators such as iNOS, ROS, IL-1 and IL-6 by diosgenin contributed to decreased production of pro-inflammatory mediators by macrophages [548]. Furthermore, the inhibition of CK2 by BMS-211 has been shown to result in a decrease of TAMs and PMN-MDSC with tumor-promoting properties in the TME, in part by interfering with myeloid cell differentiation [447]. Besides this, a role for CK2 in allergic contact dermatitis has been suggested. In this study, CK2 affected the ability of monocyte-derived DCs to mature and produce cytokines necessary to polarize effector T cells in response to chemicals related to allergic contact dermatitis [549]. In addition, Reverendo et al. elucidated an essential role for CK2 in DC functionality in relation to naive T-cell priming. A strong elevation of RNA polymerase III could be measured after CK2 phosphorylation of MAF1 homolog negative regulator of Poll III (MAF1), and its subsequent exit from the nucleus. The phosphorylation of MAF1 facilitated the translation of DC mRNAs coding for a number of co-stimulatory molecules and pro-inflammatory cytokines [550].

CK2 has been also described to target immune checkpoints such as PD-1 and PD-L1 which have gained significant importance within the last years. Recently, it has been demonstrated that phosphorylation of PD-L1 at Thr285 and Thr290 by CK2 disrupted PD-L1 binding with speckle-type POZ protein and protected PD-L1 from cullin 3 ubiquitin E3 ligase complex-mediated proteasomal degradation in cancer and DC [322]. Inhibition of CK2 decreased PD-L1 protein levels by promoting its degradation and resulted in the release of CD80 from DCs to reactivate T-cell function. Combined treatment with a CK2 inhibitor and an antibody against T-cell immunoglobulin mucin-3 (Tim-3) suppressed tumor growth and prolong survival in vivo. This suggests a new mechanism by which PD-L1 is regulated and blocking the CK2-PD-L1 pathway by inhibition of Tim-3 might be a potential anti-tumor treatment option to activate DC function [322].

#### 2.8.2. CK2 and the Adaptive Immune System

CK2 has been shown to be involved in the adaptive immune system, serving as an intrinsic regulator of effector CD4+ T-cell responses. The cytoplasmic tail of CD5, a transmembrane receptor that regulates a number of T-cell functions, contains a CK2 binding/activation domain (CD5-CK2BD) [551]. In a pre-clinical T-cell-dependent model of Multiple Sclerosis (MS) the disruption of CD5-CK2 signaling diminished the development of experimental autoimmune encephalomyelitis (EAE). This signaling has been shown to be critical for the development of CD4+ Th2 and Th17 cells, but not CD4+ Th1 cells, by regulating the threshold for T-cell anergy as well as cytokine production [552,553]. A murine model in which the regulatory subunit CK2β was specifically deleted in CD4+ Treg cells revealed that CK2β was required for a specific population of Tregs to suppress CD4+ Th2-cell-mediated allergic immune responses in the lung [554]. Upon T-cell stimulation, a significant increase in the expression of CK2 subunits, especially CK2α, could be detected, which was consistent with an increase in overall CK2 kinase activity. This suggested that activated CD4+ T-cells might utilize CK2 to support effector function [555]. Th17 cells in general are pro-inflammatory, while Tregs exhibit immunosuppressive functions. The Th17/Treg axis is relevant to many autoimmune inflammatory disorders, including MS [556]. Different groups found that inhibition of CK2 kinase activity by CX-4945 promoted the differentiation of Treg cells at the expense of Th17 cells in conjunction with suppression of PI3K/AKT/mTOR activation and STAT3 phosphorylation [555,557]. The treatment with CX-4945 ameliorated the severity of EAE due to the suppression of pathogenic Th17 cells and expansion of Treg cells [555,557]. Taken together, these results suggest that both the catalytic activity of CK2α and CK2α′ as well as CK2β-mediated regulatory mechanisms are important for the stimulation of Th17-promoting signaling pathways during CD4+ T cell activation and lineage.

The development of colitis-associated cancer is a major complication for patients with Crohn’s Disease [558]. Yang et al. demonstrated in a T-cell induced pre-clinical model for Crohn’s Disease, that the absence of CK2α in CD4+ T-cells resulted in significantly less colitis disease severity and intestinal inflammation [559]. Loss of CK2α was associated with decreased CD4+ T-cell proliferation as well as Th1 and Th17 cell responses. These findings indicate that CK2α contributes to the pathogenesis of colitis and could be explored as a treatment option for patients with Crohn’s Disease [559].

Recently, Wei et al. used a CD19-driven Cre recombinase to generate a mouse model lacking CK2α specifically in B-cells [560]. CK2α deficiency in this model resulted in an expansion of marginal zone B-cells and a reduction of transitional B-cells in the murine spleen, identifying a role for CK2α in normal B-cell development and differentiation. In addition, the group detected very low expression of all CK2 subunits and kinase activity in naïve B-cells albeit inducible upon different stimulation conditions [560].

Furthermore, an in vitro model of hepatocellular carcinoma and HeLa cells addressed whether natural killer (NK) cell-mediated cancer cell death is affected by CK2 [561]. Combinatory treatment of HepG2, Hep3B, and HeLa cells with a soluble form of recombinant TRAIL or the inhibition of CK2 by an agonistic antibody of Fas and emodin led to an increased apoptotic cell death of those cells. This seemed to be correlated with the expression levels of death receptors on the cancer cell surface. The inhibition of CK2 simultaneously increased the NK cell-mediated cell killing which was granule-independent. This implicates that CK2 inhibitors might be useful drugs to increase host tumor immunity by enhancement of cytotoxicity of NK cells [561].

An influence of CK2 on the immune system via the NF-κB signaling pathway can also be assumed. The NF-κB signaling pathway is involved in the survival, activation, and differentiation of innate and adaptive immune cells [562] and indispensable for inflammatory responses and plays a critical role in the development and progression of cancers [563]. CK2α has been shown to promote the phosphorylation of IκBα, an NF-κB inhibitor [37,56] which causes the degradation of IκBα, resulting in the release of p65 and/or p50 and translocation into the nucleus [56]. Moreover, CK2 has been described to phosphorylate the NF-κB p65 subunit on Ser529, which enhances its transcriptional activity [43,56]. It is postulated, that the inhibition of NF-κB through CK2 inhibition enables indirect tumor targeting by shifting macrophages from the tumor-tolerating M2-polarization stage towards the tumor-attacking M1-stage. In this context, Hagemann et al. demonstrated that NF-κB maintains the immunosuppressive phenotype of TAMs [564]. Combinatorial inhibition of CK2 and NF-κB seems to have the potential to “re-educate” the tumor-promoting macrophage population [564].

Beyond that, CK2 influences the Wnt/β-catenin signaling pathway, which directly alters several regulators critical for the antitumor activities of T cells. CK2 protects β-catenin from degradation by phosphorylation in the armadillo repeat protein interaction domain which leads to the induction of Wnt/β-catenin signaling, arresting CD8+ T cell development into effector cells [25,565].

Finally, CK2 can also target the transcription factor Ikaros, which is critical for lymphocyte development, especially T cells [566]. CK2 phosphorylation of Ikaros impairs its DNA binding ability, alters its subcellular localization, and leads to its ubiquitin-mediated proteasomal degradation via phosphorylation in PEST regions. Dephosphorylation of Ikaros by PP1 maintains Ikaros stability and activity [180]. In a study investigating pancreatic cancer in a murine model, it has been demonstrated that downregulation of Ikaros occurs in those pancreatic tumor-bearing mice [567]. A deregulation in the balance between CK2 and PP1 was observed, which led to the conclusion that increased CK2 activity was responsible for regulating Ikaros’ stability in this model. It has also been shown that the loss of Ikaros expression was associated with a significant decrease in CD4+ and CD8+ T cell percentages but increased CD4+ CD25+ Tregs in tumor-bearing mice [566]. Apigenin treatment restored protein expression of some Ikaros isoforms, which was speculated to be attributed to its moderate inhibition of CK2 activity. This partial restoration of Ikaros expression was accompanied by a significant increase in CD4+ and CD8+ T cell percentages, a reduction in Treg percentages and a higher production of IFN-γ by CD8+ T-cells [567].

Immune surveillance is an effective barrier to tumorigenesis and tumor progression. Nevertheless, there are cancer cells, which avoid the detection by the immune system and evade eradication and, as shown above, CK2 has an important role in those mechanisms, which still need to be further elucidated.

All CK2-dependent regulated proteins, which are described in this chapter and are responsible for the avoidance of immune destruction are demonstrated in Figure 8.

## 3. Conclusions

For a long time, CK2 was considered absolutely essential for survival as CK2α and CK2β knockouts in mouse embryos are lethal [9,10]. Very recently, C2C12 myoblasts harboring an N-terminally deleted form of CK2α′ and providing very low CK2 activity [568] and mice with conditional impaired CK2α expressing myeloid cells [544] could be successfully generated. This proves that there are at least cell populations that can exist with a very low amount of active CK2. It remains to be determined whether such defects have a long-term effect on these cell populations. However, this review demonstrates that CK2—especially when it is overexpressed—plays an extremely important role in cells and is involved in all pathways contributing to cancer development (see Figure 9). Therefore, it is reasonable that the inhibition of CK2 is one of the intensively investigated therapeutic approaches. The use of CK2 inhibitors, however, is challenged by their off-target effects and highly variable specificity (reviewed in [569,570,571,572,573]). Although it has been shown that the CK2-inhibitor CX-4945 inhibits several kinases besides CK2 with IC_50_ values ˂100 nM [574,575], CX-4945 and CIGB-300 are the only two inhibitors that have been tested in clinical trials so far (reviewed in [576,577]). Despite doubts, CK2 inhibition in the context of targeted tumor therapy and in combination with classical chemotherapeutic agents as well as the use of small molecule modulators could be of great benefit for improved therapy response. The diverse functionality of CK2, however, should not be underestimated when considering the systemic use of CK2 inhibitors.

Finally, it remains to be elucidated whether CK2 plays a role in ‘unlocking phenotypic plasticity’ and ‘senescent cells’ which have been published in 2022 as emerging hallmarks, as well as in enabling the characteristics ‘nonmutational epigenetic reprogramming’ and ‘polymorphic microbiomes’ [578].

## Figures and Tables

**Figure 1 biomedicines-10-01987-f001:**
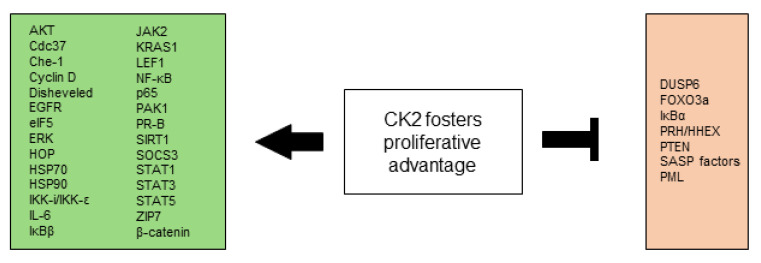
**CK2-dependent modified proteins involved in proliferative advantage.** Proteins that are activated in this process by CK2 modification are shown on the left (highlighted in green), and proteins that are restricted in their function are shown on the right (highlighted in red).

**Figure 2 biomedicines-10-01987-f002:**
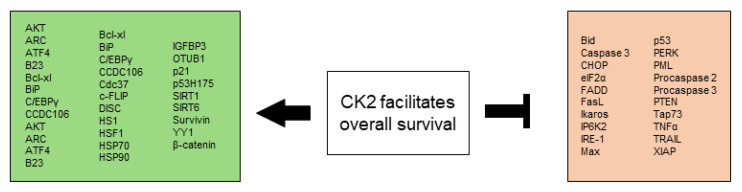
**CK2-dependent modified proteins that facilitate overall survival.** Proteins that are activated in this process by CK2 modification are shown on the left (highlighted in green), and proteins that are restricted in their function are shown on the right (highlighted in red).

**Figure 3 biomedicines-10-01987-f003:**
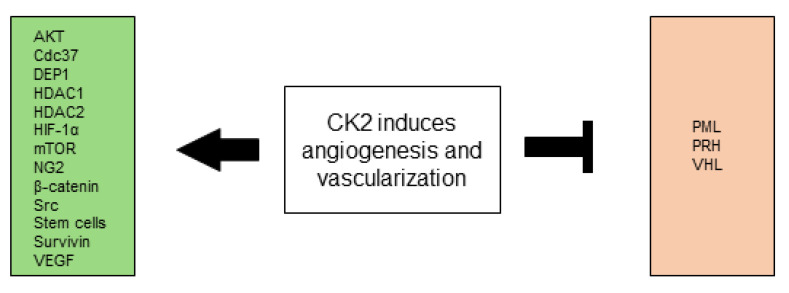
**CK2-dependent modified proteins involved in angiogenesis and vascularization.** Proteins that are activated in this process by CK2 modification are shown on the left (highlighted in green), and proteins that are restricted in their function are shown on the right (highlighted in red).

**Figure 4 biomedicines-10-01987-f004:**
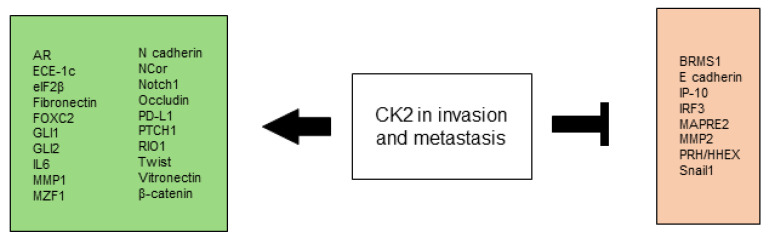
**CK2-dependent modified proteins involved in invasion and metastasis.** Proteins that are activated in this process by CK2 modification are shown on the left (highlighted in green), and proteins that are restricted in their function are shown on the right (highlighted in red).

**Figure 5 biomedicines-10-01987-f005:**
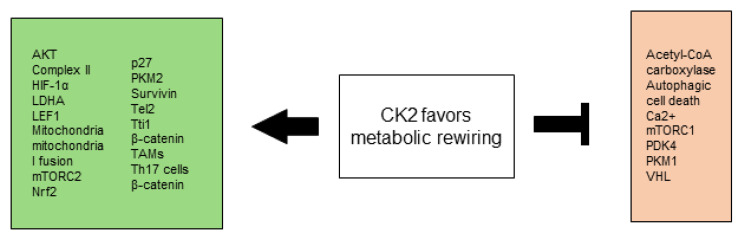
**CK2-dependent modified proteins involved in metabolic rewiring.** Proteins that are activated in this process by CK2 modification are shown on the left (highlighted in green), and proteins that are restricted in their function are shown on the right (highlighted in red).

**Figure 6 biomedicines-10-01987-f006:**
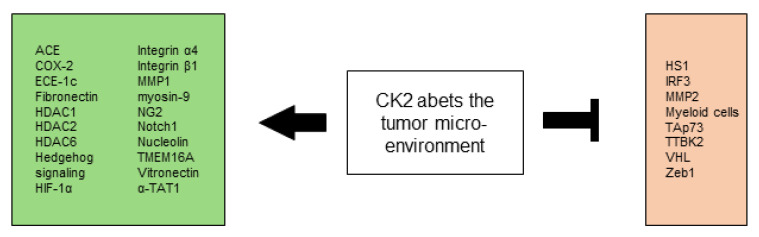
**CK2-dependent modified proteins involved in the tumor microenvironment.** Proteins that are activated in this process by CK2 modification are shown on the left (highlighted in green), and proteins that are restricted in their function are shown on the right (highlighted in red).

**Figure 7 biomedicines-10-01987-f007:**
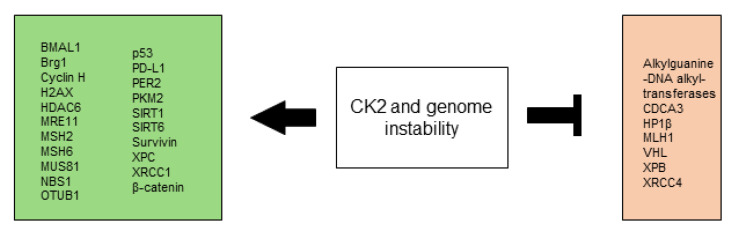
**CK2-dependent modified proteins involved in genome instability.** Proteins that are activated in this process by CK2 modification are shown on the left (highlighted in green), and proteins that are restricted in their function are shown on the right (highlighted in red).

**Figure 8 biomedicines-10-01987-f008:**
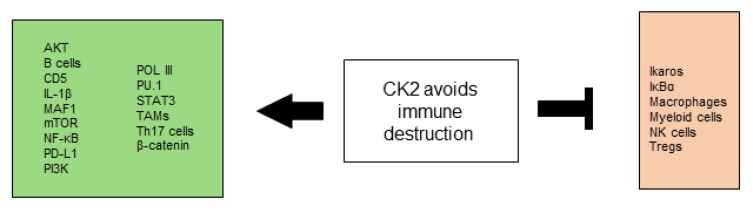
**CK2-dependent modified proteins involved in the avoidance of immune destruction.** Proteins that are activated in this process by CK2 modification are shown on the left (highlighted in green), and proteins that are restricted in their function are shown on the right (highlighted in red).

**Figure 9 biomedicines-10-01987-f009:**
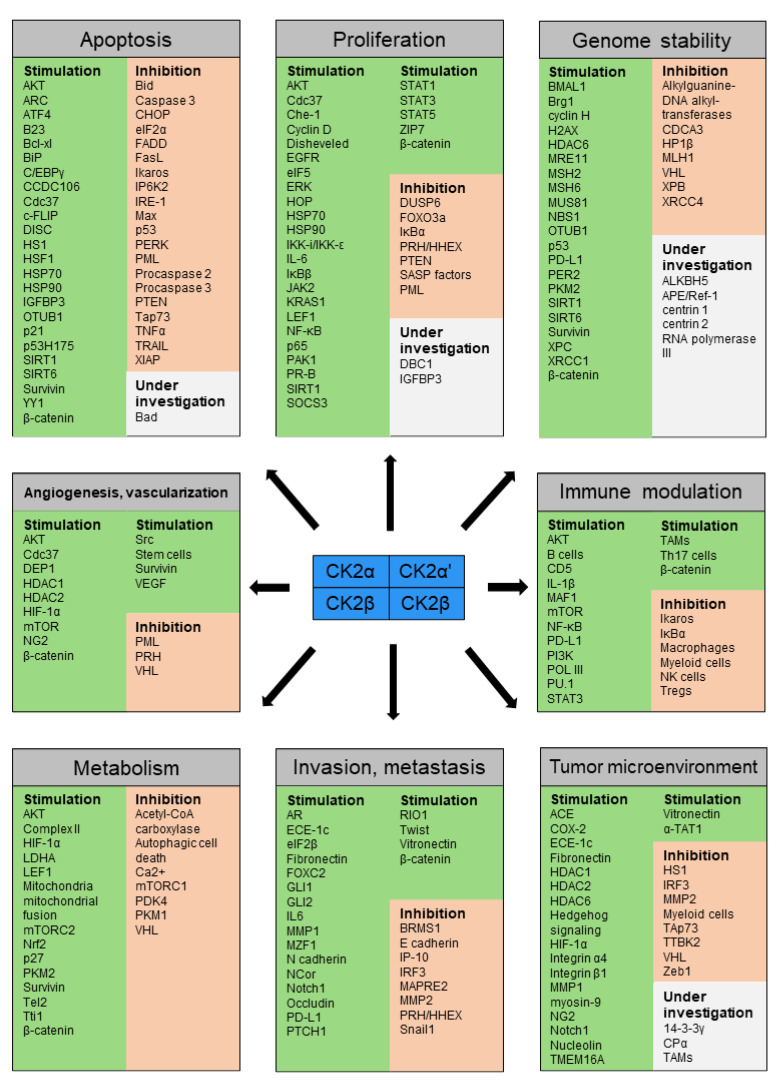
**CK2-dependent stimulation or inhibition of several proteins is responsible for cancer progression.** Shown are all cancer-associated proteins, which are phosphorylated by CK2 and mentioned in this review. Listed in the areas highlighted in red are proteins, which are functionally stimulated by this phosphorylation, listed in the areas highlighted in green, are proteins, which are functionally reduced by Ck2-dependent phosphorylation. Proteins where the effect of phosphorylation on the functionality of the protein is not yet known are highlighted in light gray.

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
