# Peer review of "CK2 and the Hallmarks of Cancer"

_biomedicines, 2022, doi:10.3390/biomedicines10081987_

Round 1
Reviewer 1 Report
The manuscript represents a comprehensive work on protein kinase CK2 and phosphorylation of proteins in connection with cancer. 561 references are impressive.
However, the manuscript in its present form has at least seven big drawbacks.
1. The huge body of information is hardly to get without illustrations. It seems impossible to address each protein mentioned in the text, but at least an overview as a figure for each chapter would be very helpful.
2. Presently, the review addresses only the phosphorylation of proteins and their role in cancer. The CK2 subunits, however, interact also with a variety of different cellular proteins and this interaction leads to altered activity of CK2 or their binding partners. This aspect is totally missing.
3. The authors describe a number of results, which were obtained by the use of different inhibitors of CK2 (DRB, TBB, CX-4945, emodin, TBBt, TBBz, and DMAT). It is known that all of these inhibitors have off target effects and their specificity is rather questionable. The authors must address this point in a revised version.
4. The Pinna group has published several papers that life without CK2a, CK2a’ and CK2b is possible, which argues against an important role of CK2 in cancer. The authors must address this point in the revised version.
5. Following the Hanahan and Weinberg line of hallmarks of cancer is an interesting idea. It leads, however, to some redundancy. This is obvious at least in lines 1876- 1883, but also in cases were CK2 phosphorylation of signalling molecules are mentioned.
6. At least 43 references are incomplete. (26, 52, 58, 75,, 86, 122, 130, 134, 143, 149, 167, 178, 180, 202, 212, 218, 225, 238, 245, 257, 263, 276, 277, 282, 289, 292, 303, 311, 384, 418, 448, 466, 511, 536, 540, 543, 550, 551, 552, 553, 560, 561). Moreover, the format of the references is not uniform. The same journal is written in different ways (PLOS ONE, plos one, Plos ONE etc.). Is “Nature Publishing group” a journal? Reference 556.
7. Line 1725 How do the authors explain the difference between in vitro and in vivo phosphorylation sites. Did the authors use proteins from different species? Zschoernig and Mahlknecht have shown that ser659 and ser661 are phosphorylated by CK2 in vivo and in vitro. Reference 206 cited a paper about SIRT1 phosphorylation by CK2 after ionizing radiation as S154, S649, S651, and S689. (Kang H, Jung JW, Kim MK, Chung JH. CK2 is the regulator of SIRT1 substrate-binding affinity, deacetylase activity and cellular response to DNA-damage. PLoS One. 2009;4(8):e6611). Thus, the authors should check their statements about CK2 and SIRT1 phosphorylation. Probably, there are more inconsistencies in citing the literature. I just made a random sample check. The authors did a great job but they should take some time to check their statements and the cited literature carefully.
Minor points:
Line 199 must read: ….the phosphorylation of Akt on ser129 by CK2 promotes Akt kinase activity.
Line 416 (PARP) is redundant
Line 418 the authors should mention that the Litchfield group has shown that CK2 phosphorylates caspase-3 to prevent its activation by upstream caspases.
Line 504 the first papers about CK2 binding to p21 have been published already in the years 1996- 2000 by Götz et al. It was further shown that p21 inhibits the CK2 kinase activity.
Line 514 Romero-Oliva and Allende have shown that p21 is phosphorylated by CK2 at least in vitro.
Line 602 The first papers on CK2 binding to p53 and the phosphorylation of p53 by CK2 were published between 1994-1997
Line 711 another UPR signalling namely CREB3, also known as luman, was recently published as a CK2 substrate. CK2 phosphorylation leads to destabilization of CREB3
Line 875 the authors should check the meaning of “vasatation”.
2.4.6 unbalanced expression of CK2 subunits. A statement whether the unbalanced expression of CK2 subunits has an influence on the kinase activity of CK2 or does it have an influence interaction of CK2 with other cellular proteins or both.
Line 1116 et al. should be written in italics as in other cases.
Author Response
Manuscript #biomedicines-1785589
Thank you for all comments on our review and the opportunity to submit a revised version.
Answers to Reviewer#1’s comments:
- Reviewer#1 remarked that the “huge body of information is hardly to get without illustrations. It seems impossible to address each protein mentioned in the text, but at least an overview as a figure for each chapter would be very helpful.”
We appreciate this idea and added in the end of each chapter one figure (see Figure 1-8) to summarize all proteins, which are demonstrated to be involved.
- Reviewer#1 mentioned “the review addresses only the phosphorylation of proteins and their role in cancer. The CK2 subunits, however, interact also with a variety of different cellular proteins and this interaction leads to altered activity of CK2 or their binding partners. This aspect is totally missing.”
We completely agree that we only refer in this review a part of the functional diversity of CK2.
To clarify this, we added the following section to the introduction:
“It is important to acknowledge that CK2 phosphorylation sites have been shown to overlap with other posttranslational modifications (reviewed in [1]), thus participating in numerous regulatory events. Furthermore, it is known that CK2 activity is regulated in various biological processes and that CK2 interacts with different cellular proteins, leading to altered activity of CK2 as well as its partners (reviewed in [2]). However, this review only focuses on CK2-mediated phosphorylation of proteins and their role in cancer.”
- Reviewer#1 remarked that a number of results were described,“which were obtained by the use of different inhibitors of CK2 (DRB, TBB, CX-4945, emodin, TBBt, TBBz, and DMAT). It is known that all of these inhibitors have off target effects and their specificity is rather questionable. The authors must address this point in a revised version. “
To clarify this, we included a comment in the conclusion section and added several reviews that explicitly address CK2 inhibition or clinical trials that use CK2 inhibitors.
- Reviewer#1 pointed out that “several papers demonstrated that life without CK2a, CK2a’ and CK2b is possible, which argues against an important role of CK2 in cancer. The authors must address this point in the revised version.”
We thank reviewer#1 for mentioning this important point and added in our conclusion:
“CK2 is essential for survival as CK2α and CK2β knockouts in mouse embryos are lethal [3], [4] while CK2α′ knockouts are viable but lead to male infertility [5]. Since CK2 is involved with a plethora of biological processes (reviewed in [1], [2]) it is an obvious assumption that CK2 plays an important role in cancer (recently reviewed in [6]).“
- Reviewer#1 remarked “Following the Hanahan and Weinberg line of hallmarks of cancer is an interesting idea. It leads, however, to some redundancy. This is obvious at least in lines 1876- 1883, but also in cases were CK2 phosphorylation of signalling molecules are mentioned. “
We agree that there are overlaps. However, it was and still is important to list the individual proteins repeatedly, as they are important in all the contexts mentioned.
Since Hanahan has published 2022 a new review about “hallmarks of cancer” we included the following sentence in our conclusion:
“It remains to be elucidated whether CK2 plays a role in ‘unlocking phenotypic plasticity’ and ‘senescent cells’ which have been published in 2022 as emerging hallmarks as well as in the enabling characteristics ‘nonmutational epigenetic reprogramming’ and ‘polymorpic microbiomes’ [7].”
- Reviewer#1 pointed out that “the format of the references is not uniform”.
We checked the format of all references and set all wrong listed references to the correct format.
- Reviewer#1 remarked that “the authors should check their statements about CK2 and SIRT1 phosphorylation”
We corrected our statement.
- In addition, reviewer #1 found several minor points: a) Line 199 must read: ….the phosphorylation of Akt on ser129 by CK2 promotes Akt kinase activity.
We changed this.
b) Line 416 (PARP) is redundant.
We corrected this.
c) Line 418 the authors should mention that the Litchfield group has shown that CK2 phosphorylates caspase-3 to prevent its activation by upstream caspases.
We included this.
d) Line 504 the first papers about CK2 binding to p21 have been published already in the years 1996- 2000 by Götz et al. It was further shown that p21 inhibits the CK2 kinase activity.
We added Götz et al. and mentioned the influence of p21 on CK2
e) Line 514 Romero-Oliva and Allende have shown that p21 is phosphorylated by CK2 at least in vitro.
That’s right. We already mentioned this in our review line 521/522
f) Line 602 The first papers on CK2 binding to p53 and the phosphorylation of p53 by CK2 were published between 1994-1997
We included the original paper and removed the mentioned review.
g) Line 711 another UPR signalling namely CREB3, also known as luman, was recently published as a CK2 substrate. CK2 phosphorylation leads to destabilization of CREB3
We added CREB3 in chapter 2.2.2.3, in the abbreviations and in our figures.
h) Line 875 the authors should check the meaning of “vasatation”.
We corrected this.
i) 2.4.6 unbalanced expression of CK2 subunits. A statement whether the unbalanced expression of CK2 subunits has an influence on the kinase activity of CK2 or does it have an influence interaction of CK2 with other cellular proteins or both.
We added one sentence to the end of chapter 2.4.6.
j) Line 1116 et al. should be written in italics as in other cases.
We changed this.
Reviewer 2 Report
This review systemically summarizes current evidence on the functional importance of CK2 in regulating different hallmarks of cancer. The paper is important to those with related research interests; however, some contents seem irrelevant to the section title, e.g., section 2.8 (CK2 is responsible for immune modulation) and should be re-organized carefully.
1. Usually a review article should be started with a section on general type of issue and then narrowed down to some specific issues that are most correlated with the research question or statement. However, this is not the case for this study. Many sub-sections need to be revised to support the section title (section 2.1 (2.1.2); section 2.4 (2.4.2, 2.4.3, 2.4.4 and 2.4.6); section 2.6 (2.6.3)).
2. section 2.1, CK2 is involved in selective growth and proliferative advantage. I am confused about the difference between growth and proliferation. The authors may elaborate on this.
3. line 292, “Its main role is in assisting other proteins in folding properly, stabilizing proteins against heat stress, and promoting protein degradation.” Contradictory role of CK2 on protein stability?
4. The authors may also provide a table summarizing small molecule drugs targeting CK2 that are being successfully used in cancer treatment.
Author Response
Manuscript #biomedicines-1785589
Thank you for all comments on our review and the opportunity to submit a revised version.
Answers to Reviewer#2’s comments:
1. Reviewer#2 pointed out that “some contents seem irrelevant to the section title, e.g., section 2.8 (CK2 is responsible for immune modulation) and should be re-organized carefully”.
The hallmark “Avoiding immune destruction” is newly included in the hallmarks of cancer since 2022. Therefore, we would like to leave a paragraph about the immune system in our review.
However, to go in line with the exact name of this new hallmark of cancer, we changed the headline of section 2.8 into “CK2 contributes to the avoidance of immune destruction” and expanded the introduction in chapter 2.8:
“The main function of the mammalian immune system is to protect against invading or infectious pathogens and to eliminate damaged cells while monitoring tissue homeostasis. The relationship between immune cells and developing tumors is complex as cancer cells employ various mechanisms mimicking peripheral immune tolerance thereby avoiding detection and elimination [546]. Recent studies showed that CK2 is an important factor for normal physiological function of immune cells but also of significant relevance in the modulation of inflammation [547] therefore, it is reasonable to assume that CK2's involvement with the immune system also contributes to tumorigenesis. However, the roles of CK2 in the regulation of immune cells are incompletely understood so far.“
2. Reviewer#2 remarked that “Many sub-sections need to be revised to support the section title (section 2.1 (2.1.2); section 2.4 (2.4.2, 2.4.3, 2.4.4 and 2.4.6); section 2.6 (2.6.3).”
To make this clearer, we included introductory sentences to all mentioned sections to better link them to the main heading.
3. Reviewer#2 criticizes the headline of section 2.1 “CK2 is involved in selective growth and proliferative advantage” and is confused about the difference between growth and proliferation. The authors may elaborate on this.”
In our review, we based our listing of hallmarks of cancer on the literature. "Selective growth and proliferative advantage" is defined in the literature as one hallmark of cancer.
In principle, we agree with the reviewer that "growth" and "proliferation" are synonymous terms.
We have also used these terms synonymous in our review. Nevertheless, we wanted to include all hallmarks of cancer, which are commonly used in the literature, in our reviews.
4. Reviewer#2 wishes to clarify the message of one sentence in line 292, “Its main role is in assisting other proteins in folding properly, stabilizing proteins against heat stress, and promoting protein degradation.”
Therefore, we changed „it’s“(line 306) to „HSP90‘s“ to clarify the relationship of the statements to each other.
5. Reviewer#2 would appreciate it if a table summarizing small molecule drugs targeting CK2 that are being successfully used in cancer treatment would be included in this review.
So far, only CX-4945 and CIGB-300 are being tested in clinical trials. However, there is a review on small molecules targeting CK2 (Small molecule modulators targeting protein kinase CK1 and CK2) and a recent review on CK2 inhibition (Chapter Two - Protein kinase CK2 inhibition as a pharmacological strategy). Both are now mentioned in the conclusion (in the context of CK2 inhibition) in the revised version of our review.
Reviewer 3 Report
The authors have submitted a very detailed and elaborate review article on Casein kinase 2. The paper itself is well written and should be highly evaluated. However, it is very long for a review submitted to a journal and contains a large number of references. The reviewer cannot determine whether or not this lengthy review is consistent with the policy and intent of the journal. It might be a good idea to shorten the paper a bit. I believe that the editorial board should decide on its acceptance.
Author Response
Manuscript #biomedicines-1785589
Thank you for all comments on our review and the opportunity to submit a revised version.
Answer to Reviewer#3’s comments:
Reveiwer#3 pointed out that “The authors have submitted a very detailed and elaborate review article on Casein kinase 2. The paper itself is well written and should be highly evaluated. ... I believe that the editorial board should decide on its acceptance.”
We thank rewiever#3 very much for this positive feedback.
Round 2
Reviewer 1 Report
The revised version of the manuscript has improved. The insertion of the figures at the end of each chapter is appreciated. Schematic drawings, however, would be more impressive than just listing the proteins. The references are now listed properly.
The authors have inserted a statement about the off-target effects of CK2 inhibitors. However, a more critical view on the use of CK2 inhibitors is urgently necessary. Two recent reviews on the specificity of SGC-CK2-1 and CX-4945 are still missing (Salvi et al. 2021, Cell Death Discov. 7, 325; Licciardello MP, Workman P. Trends Pharmacol. Sci. 2021, 42(5):313-315). CX-4945 has often been used in clinical trials and is mentioned in the manuscript several times. It is, however, known that CX-4945 significantly inhibits several kinases with IC50 values ˂100 nM (Pierre et al. 2011, J. Med. Chem 54, 635-654, Wells et al. 2021, Cell Chem. Biol. 28, 546- 558).
Point 4 of the previous review is not answered properly in the revised version of the manuscript. Knock-out experiments for all three CK2 subunits have been reported by the Pinna group and conditional knock-out experiments by Larson have shown that life without CK2 is possible. This is a critical point for a manuscript addressing the role of CK2 in cancer. Thus, the authors should address this point.
The newly inserted sentence in lines 1156 ff is incorrect. An imbalanced expression of CK2 subunits has an impact on the kinase activity. There are several reports showing that the substrate specificity differs for CK2a, CK2a‘, and the CK2 holoenzyme. Furthermore, CK2b is known to regulate the substrate specificity of CK2a. These observations have led the Pinna group to define 3 classes of CK2 substrates. Class I substrates, which are phosphorylated by CK2a and the holoenzyme, class II substrates, which are by CK2a alone and class III substrates, which are phosphorylated by the CK2 holoenzyme alone. (Pinna, J. Cell. Sci. 115, 3873- 3878). Now, it is known that there are also CK2a‘ specific substrates. These different specificities are also a problem for CK2 inhibition. Some inhibitors inhibit CK2a, CKa‘, and the holoenzyme, or they inhibit only the CK2 holoenzyme because they inhibit the assembly of the holoenzyme.
By reading the manuscript I noticed some minor mistakes.
Line 260: ….in in vitro as well as in vivo in models…..
Line 774 must read: …..cAMP responsive……
Please check the sentences in lines 931-934.
Line 1372 …. et al. (italic)
Line 1713 should be: …..switch off ….
There may be more mistakes.
Author Response
Answers to the second comments of reviewer#1
1. The revised version of the manuscript has improved. The insertion of the figures at the end of each chapter is appreciated. Schematic drawings, however, would be more impressive than just listing the proteins. The references are now listed properly.
We are pleased that the reviewer approves the revision. Schematic drawings would be definitively more attractive, but it also runs the risk of making the relationships unclear. We would like to keep the listing.
2. The authors have inserted a statement about the off-target effects of CK2 inhibitors. However, a more critical view on the use of CK2 inhibitors is urgently necessary.
Two recent reviews on the specificity of SGC-CK2-1 and CX-4945 are still missing (Salvi et al. 2021, Cell Death Discov. 7, 325; Licciardello MP, Workman P. Trends Pharmacol. Sci. 2021, 42(5):313-315). CX-4945 has often been used in clinical trials and is mentioned in the manuscript several times. It is, however, known that CX-4945 significantly inhibits several kinases with IC50 values ˂100 nM (Pierre et al. 2011, J. Med. Chem 54, 635-654, Wells et al. 2021, Cell Chem. Biol. 28, 546- 558).
We agree with reviewer#1 that off-target effects of inhibitors are important to discuss. To emphasize this even more, we added the two missing reviews (Salvi et al. 2021, Cell Death Discov. 7, 325; Licciardello MP, Workman P. Trends Pharmacol. Sci. 2021, 42(5):313-315) to those we already cited reviews concerning off-target effects of CK2-inhibitors in our conclusion section. Furthermore, we specified our statement concerning the off-target effect of CX-4945 by adding:
"Although it has been shown that the CK2-inhibitor CX-4945 inhibits several kinases besides CK2 with IC50 values ˂100 nM [588,589], CX-4945 and CIGB-300 are the only two inhibitors which are tested in clinical trials so far (reviewed in [590,591])."
3. Point 4 of the previous review is not answered properly in the revised version of the manuscript. Knock-out experiments for all three CK2 subunits have been reported by the Pinna group and conditional knock-out experiments by Larson have shown that life without CK2 is possible. This is a critical point for a manuscript addressing the role of CK2 in cancer. Thus, the authors should address this point.
To address this point in more detail, we have significantly revised the statement of our conclusion section:
"For a long time, CK2 was considered as absolutely essential for survival as CK2α and CK2β knockouts in mouse embryos are lethal [580,581]. Very recently, viable CK2α/α'−/− C2C12 myoblasts [582] and mice harboring conditional impaired CK2α expressing myeloid cells [555] could be successfully generated. This proves that there are cell populations which are able to exist without functional CK2. It remains to be determined whether such a defect has a long-term effect on these cell populations. However, this review demonstrates that CK2 – especially when it is overexpressed - plays an extremely important role in cells and is involved in all pathways contributing to cancer development (see Figure 9). Therefore, it is reasonable that inhibition of CK2 is one of the intensively investigated therapeutic approaches. The use of CK2 inhibitors, however, is challenged by their off-target effects and highly variable specificity (reviewed in [583–587]). Although it has been shown that the CK2-inhibitor CX-4945 inhibits several kinases besides CK2 with IC50 values Ë‚100 nM [588,589], CX-4945 and CIGB-300 are the only two inhibitors which are tested in clinical trials so far (reviewed in [590,591]). Despite doubts, CK2 inhibition in the context of targeted tumor therapy and in combination with classical chemotherapeutic agents as well as the use of small molecule modulators could be of great benefit for improved therapy response. The diverse functionality of CK2, however, should not be underestimated when considering the systemic use of CK2 inhibitors. Finally, it remains to be elucidated whether CK2 plays a role in ‘unlocking phenotypic plasticity’ and ‘senescent cells’ which have been published in 2022 as emerging hallmarks as well as in the enabling characteristics ‘nonmutational epigenetic reprogramming’ and ‘polymorpic microbiomes’ [592]."
4. The newly inserted sentence in lines 1156 ff is incorrect. An imbalanced expression of CK2 subunits has an impact on the kinase activity.
Furthermore, CK2b is known to regulate the substrate specificity of CK2a. These observations have led the Pinna group to define 3 classes of CK2 substrates. Class I substrates, which are phosphorylated by CK2a and the holoenzyme, class II substrates, which are by CK2a alone and class III substrates, which are phosphorylated by the CK2 holoenzyme alone. (Pinna, J. Cell. Sci. 115, 3873- 3878). Now, it is known that there are also CK2a‘ specific substrates.
We are totally in line with reviewer#1 that unbalanced expression of CK2 subunits has an important impact on the kinase activity and that different specificities are a problem for CK2 inhibition. We apologize that this message was not clear enough in the first revised version of our review.
To clarify our message we modified the text of paragraph 2.4.6. as following:
CK2 can function as a tetramer but its individual subunits function outside of the tetrameric complex as well [2]. Given CK2s extensive role in invasion and metastasis an unbalanced expression of CK2 subunits could further contribute to this process.
Unbalanced expression of CK2α and CK2β subunits has been described for many different mammalian tissues [350–352]. Furthermore, different substrate specificity has been detected for the CK2α and the CK2α’ subunits as well as the CK2 holoenzyme [353], leading to the definition of class I-III CK2 substrates [354]. Therefore, it becomes clear that the unequal expression of CK2 components has an important impact on the overall CK2 activity. In the context of invasion and metastasis the discrepancy in expression of CK2 kinase subunits was described to drive EMT by Snail1 induction [355].
By reading the manuscript I noticed some minor mistakes.
We corrected all points.
Reviewer 2 Report
Accept in present form.
Author Response
We thank reviewer#2 for reviewing our revised review.
Round 3
Reviewer 1 Report
The authors did a great job in collecting these data.
Author Response
We would like to thank reviewer#1 for the intensive review and the many important suggestions for improvement.